**EMBO** *reports*

# HDAC6 deacetylates TRIM56 to negatively regulate cGAS-STING-mediated type I interferon responses

Qiongzhen Zeng[1,2,8], Zixin Chen [ID] [3,8], Shan Li[4], Ziwei Huang [ID] [4], Zhe Ren[2], Cuifang Ye[2], Xiao Wang [ID] [5], Jun Zhou [ID] [6], Kaisheng Liu [ID] [1✉], Kai Zheng [ID] [7✉] & Yifei Wang [ID] [2✉]

## Abstract

**Histone deacetylase HDAC6 has been implicated in regulating antiviral innate immunity. However, its precise function in response to DNA virus infection remains elusive. Herein, we find that HDAC6 deficiency promotes the activation of cGAS-STING signaling and type I interferon (IFN) production, both in vitro and in vivo, resulting in a decrease in HSV-1 infection. Mechanistically, HDAC6 deacetylates tripartite motif protein 56 (TRIM56) at K110 in mice, thereby impairing the monoubiquitination cGAS and its DNA binding ability. Overexpression of TRIM56 K110Q protects mice against HSV-1 infection. Notably, different amino acids at position 110 of TRIM56 in human and mouse cause species-specific IFN responses. Further, we show that during early stages of HSV-1 infection, the viral protein US3 phosphorylates HDAC6 to inhibit the cGAS-mediated antiviral response. Our results suggest that HDAC6 inhibits cGAS activation through TRIM56 deacetylation in a species-specific manner.**

**Keywords** HSV-1; HDAC6; TRIM56; cGAS; Interferon
**Subject Categories** Immunology; Post-translational Modifications & Proteolysis; Signal Transduction

## Introduction

HSV-1 is a neurotropic DNA virus and member of the alphaherpesvirus subfamily. It is a widely prevalent human pathogen, primarily transmitted through close contact between infected and susceptible individuals, resulting in labial infection (Marcocci et al, 2020). HSV-1 virions gain entry to epithelial cells by fusing with the plasma membrane envelope. Subsequently, the envelope protein and capsid, which contain the linear viral genome, are released into the cytoplasm. Subsequently, the capsid is transported via microtubules to the nuclear pore, where the viral genome is released into the nucleus. Subsequently, following replication in epithelial cells, HSV-1 capsids undergo retrograde axonal transport to reach ganglion neurons, where they release viral DNA into the nucleus, thereby establishing latency. A lytic infection or reactivation from latency of HSV-1 can result in the development of herpes labialis, keratitis, and herpes simplex encephalitis (HSE), which may lead to lifelong recurrent clinical or subclinical episodes (Marcocci et al, 2020).

The type I interferon (IFN) response plays a crucial role in regulating viral infection and cancer immunotherapy (Snell et al, 2017). Cyclic GMP-AMP (cGAMP) synthase (cGAS) acts as a cytoplasmic DNA sensor, detecting viral or self double-stranded DNA, and producing the second messenger cGAMP. This in turn activates STING and subsequently recruits TANK-binding kinase 1 (TBK1), resulting in the activation of interferon regulated factor 3 (IRF3). This leads to the induction of type I IFN expression and resulting antiviral or anticancer immune responses (Margolis et al, 2017; Reinert et al, 2016). Alternatively, continuous activation of cGAS-STING triggers the nuclear factor kappa B (NK-κB) signaling pathway to release numerous inflammatory factors (Gulen et al, 2023; Reinert et al, 2021). Different posttranslational modifications, such as ubiquitination and phosphorylation, have been extensively demonstrated to regulate the cGAS-STING axis (Liu et al, 2022; Yu et al, 2022). Thus, precise regulation of the activation of the cGAS-STING signaling pathway is essential to maintain immune homeostasis.

Tripartite motif (TRIM) protein 56 (TRIM56), also known as ring finger protein 109 (RNF109), has a RING domain exhibiting deubiquitinase activity, a B-box domain with unknown function, and a unique C-terminal region that shows little similarity to other TRIM C-terminal regions (Fu et al, 2023). Importantly, TRIM56 plays a crucial role in antiviral defense by either directly inhibiting viral infection or regulating the host antiviral type I IFN response through modulating the toll-like receptor (TLR) and cGAS-STING signaling pathways (Fu et al, 2023). For instance,

[1]Guangdong Provincial Clinical Research Center for Geriatrics, Shenzhen Clinical Research Center for Geriatrics, Shenzhen People's Hospital (The Second Clinical Medical College, Jinan University), 518020 Shenzhen, China. [2]Institute of Biomedicine, College of Life Science and Technology, Guangdong Province Key Laboratory of Bioengineering Medicine, Key Laboratory of Innovative Technology Research on Natural Products and Cosmetics Raw Materials, Jinan University, 510632 Guangzhou, China. [3]Guangdong Cardiovascular Institute, Medical Research Institute, Guangdong Provincial Geriatrics Institute, Guangdong Provincial People's Hospital (Guangdong Academy of Medical Sciences), Southern Medical University, 510080 Guangzhou, China. [4]Guangdong Provincial Engineering Center of Topical Precise Drug Delivery System, School of Pharmacy, Guangdong Pharmaceutical University, 510006 Guangzhou, China. [5]Department of Pharmacy, Shenzhen People's Hospital (The Second Clinical Medical College, Jinan University), 518020 Shenzhen, China. [6]State Key Laboratory of Medicinal Chemical Biology, College of Life Sciences, Nankai University, 300071 Tianjin, China. [7]School of Pharmacy, Shenzhen University Medical School, Shenzhen University, 518055 Shenzhen, China. [8]These authors contributed equally: Qiongzhen Zeng, Zixin Chen.
✉E-mail: liukaisheng@szhospital.com; zhengk@szu.edu.cn; twangyf@jnu.edu.cn

TRIM56 stimulates K63-linked polyubiquitination of STING at K150, thereby enhancing the formation of STING dimers or oligomers to recruit TBK1 (Tsuchida et al, 2010; Yang et al, 2018). TRIM56 also induces monoubiquitination of cGAS at K335, leading to a significant increase in its dimerization, DNA-binding activity, and cGAMP production (Seo et al, 2018). Nevertheless, studies have indicated that TRIM56 is capable of impeding DNA viruses through its RING and C-terminal domains (Tian et al, 2022); however, the function of the B-box domain remains uncertain and further research is needed to elucidate the regulatory mechanism of TRIM56.

Histone deacetylase 6 (HDAC6) is a member of the HDAC family II that primarily catalyzes the deacetylation of cytoplasmic nonhistone proteins, thus playing a critical role in viral infection and antiviral immune response (Qu et al, 2023; Zheng et al, 2017). For example, HDAC6 suppresses poly (I:C)-induced IFN-β production via modification of TBK1 enzymatic activity (Wang et al, 2020). HDAC6 activates cytoplasmic retinoic acid-inducible gene I-like receptor (RIG-I) to enhance IFN transcription and improve the antiviral immune response (Choi et al, 2016; Liu et al, 2016). In addition, the deacetylase activity of HDAC6 regulates TRIM22 transcription, regulating IFN-mediated antiviral activity (Gao et al, 2013). Although there has been extensive research on the role of HDAC6 in regulating IFN, its regulation of the cGAS-mediated immune response to DNA viruses remains unclear.

In this study, we investigated the role of HDAC6-mediated antiviral immune response during HSV-1 infection. We found that HDAC6 inhibits the intracellular DNA-mediated IFN response, thereby promoting HSV-1 infection in neuronal cells. Consistent with the in vitro results, HDAC6 knockout (KO) mice were resistant to HSV-1 infection and herpes simplex encephalitis (HSE) pathogenesis. Through LC‒MS/MS-based HDAC6-Interactome and Acetylome, we revealed that HDAC6 directly binds to and deacetylates TRIM56 at K110 within the B-box1 domain, thereby hindering cGAS-STING-mediated IFN induction. Acetylation of TRIM56 K110 was key to its binding to HDAC6 and regulating the IFN response, as evidenced by the distinct effects of TRIM56 K110Q and K110R overexpression on the cGAS-STING axis. Interestingly, TRIM56 displayed species-specific differences in amino acid composition at position110, which leads to a differentially regulated IFN response in human and mouse. Furthermore, mice overexpressing TRIM56 WT and TRIM56 K110Q exhibited significantly increased IFN activation and reduced HSV-1 proliferation. Finally, we discovered that the viral protein US3 phosphorylates cytoplasmic HDAC6 to increase its deacetylase activity and interaction with TRIM56 in the early infection stage. In summary, our findings disclose a previously unappreciated mechanism by which HDAC6-mediated TRIM56 K110 deacetylation controls cGAS-mediated IFN responses.

# Results

## HDAC6 inhibits the intracellular DNA-triggered antiviral IFN response via its deacetylase activity

The role of HDAC6 in the antiviral IFN response to HSV-1 infection was assessed. The knockdown of HDAC6 using siRNAs significantly enhanced the expression of Ifn-β and the IFN-stimulated genes Cxcl10 and Isg15 in HSV-1-infected N2a cells when compared to the NC group (Fig. EV1A,B). Additionally, HDAC6 knockdown enhanced the activation of TBK1 and IRF3 upon HSV-1 infection, coupled with a decrease in the viral protein ICP0 (Fig. 1A). HDAC6 depletion consistently increased TBK1 and IRF3 activation in response to other cytoplasmic DNA stimuli, including calf thymus (CT) DNA and interferon stimulatory DNA (ISD; double-stranded 45-base-pair oligonucleotides lacking CpG sequences) (Fig. 1B,C). Given that si-HDAC6 siRNA #2 demonstrated the most robust effect, it was selected for subsequent experiments. HDAC6 knockdown by si-HDAC6 siRNA #2 induced time-dependent TBK1 and IRF3 activation in HSV-1 infected N2a cells (Fig. 1D) and BV2 cells (Fig. EV1C), resulting in elevated IFN-β production (Fig. 1E). Conversely, HDAC6 overexpression caused a significant decrease in α-tubulin acetylation (Fig. EV1D), prevented HSV-1 infection-induced TBK1 and IRF3 phosphorylation (Fig. 1F), and ultimately effectively inhibited IFN-β production (Fig. 1G). Therefore, these findings suggest that HDAC6 plays a negative regulatory role in intracellular DNA-mediated antiviral innate immunity.

In addition, we investigated whether HDAC6 inhibits the innate antiviral response through its deacetylase function by constructing an HDAC6 mutant that lacks deacetylase activity (HDAC6-HD1/2 m) (Zhang et al, 2006). Overexpression of HDAC6-HD1/2 m resulted in a significant increase in the acetylation of α-tubulin compared to HDAC6 WT (Fig. EV1E). It also upregulated the phosphorylation of TBK1 and IRF3 and reduced the expression of ICP0 (Fig. 1H). Additionally, HDAC6-HD1/2 m restored the production of IFN-β (Fig. 1I) and the expression of the IFN-stimulated genes Isg15, Isg20 and Mx2 (Fig. EV1F). Moreover, the inhibition of HDAC6 deacetylase activity through its specific inhibitor tubacin notably amplified the HSV-1-induced expression of Ifn-α, Ifn-β, Cxcl10 and Isg15 (Fig. EV1G,H), while also promoting the activation of TBK1 (Fig. EV1I). Therefore, these findings indicate that HDAC6 suppresses the intracellular DNA-triggered antiviral IFN response via its deacetylase activity.

## Inhibition of HDAC6 restricts HSV-1 infection in vitro

The impact of HDAC6 on the antiviral response to HSV-1 infection was investigated. Viral plaque assays revealed a significant reduction in the number of viral plaques following HDAC6 knockdown (Figs. EV2A and 2A). Additionally, HDAC6 knockdown inhibited the expression of viral titers in N2a cells (Fig. 2B). In contrast, the overexpression of HDAC6 led to an increase in viral plaque numbers (Fig. 2C). Therefore, these findings suggest that HDAC6 inhibits the antiviral response, promoting HSV-1 infection. Furthermore, the inhibition of HDAC6 deacetylase activity by HDAC6-HD1/2 m overexpression or tubacin caused a reduction in viral plaque numbers and suppressed viral gene expression in N2a cells (Fig. 2D–F), as well as EGFP-tagged HSV-1 infection (Fig. 2G). Furthermore, treatment with tubacin suppressed viral gene expression (Fig. EV2B,C) and viral plaque formation in BV2 and HMC3 microglial cells (Fig. EV2D,E). Western blot assays further indicated that tubacin inhibited the early stages of HSV-1 infection (Fig. EV2F,G), thereby supporting the finding that HSV-1 enhances HDAC6 deacetylase activity to prevent the host innate antiviral IFN response during the early infection stage.

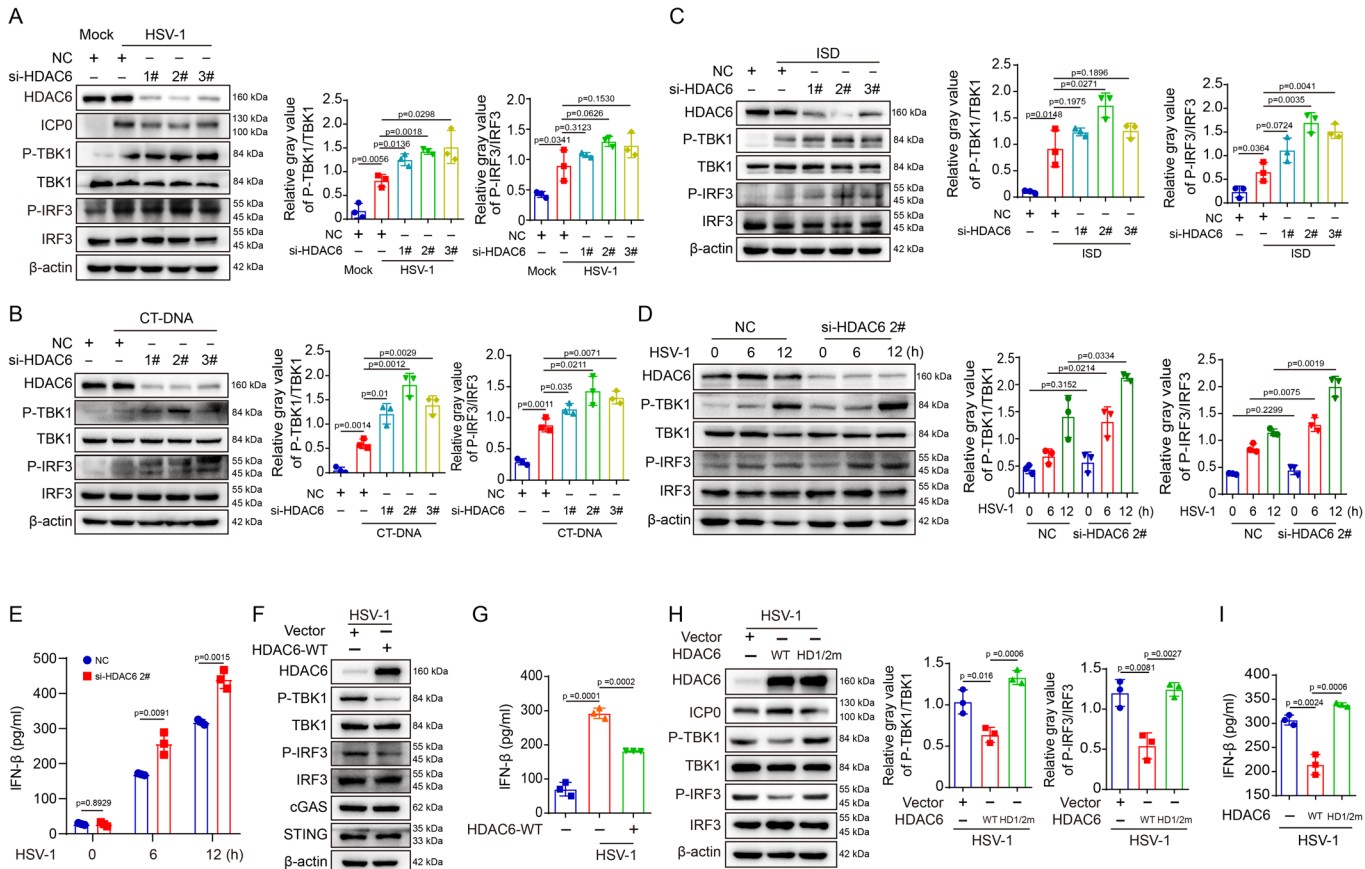

**Figure 1. HDAC6 negatively regulates IFN production.**

(A–C) N2a cells were transfected with negative control (NC) siRNA or HDAC6 siRNA for 72 h followed by infection with HSV-1 (MOI = 3) for 12 h (A), stimulated with 2 μg/ml CT-DNA (B) or 2 μg/ml ISD (C) for 6 h. Western blotting was used to assess the levels of indicated proteins in cell lysates. Histograms show the quantification of P-TBK1-1 or P-IRF3 levels relative to that of TBK1 or IRF3. Data were analyzed using the unpaired *t* test, which are shown as mean ± SD (*n* = 3 biological replicates). (D, E) N2a cells were transfected with negative control (NC) siRNA or HDAC6 siRNA for 72 h followed by infection with HSV-1 (MOI = 3) for indicated time periods. Western blot analysis of the indicated proteins in cell lysate. (D) Histograms show the quantification of P-TBK1 or P-IRF3 levels relative to that of TBK1 or IRF3. Data were analyzed using the unpaired *t* test, which are shown as mean ± SD (*n* = 3 biological replicates). (E) ELISA assay detection of IFN-β production. Data were analyzed using the unpaired *t* test, which are shown as mean ± SD (*n* = 3 biological replicates). (F–I) N2a cells were transfected with vector, HDAC6 WT or deacetylase activity mutation (HD1/2 m) for 72 h followed by infection with HSV-1 (MOI = 3) for 12 h. The levels of proteins involved in the cGAS-STING signaling were analyzed by western blot (F, H) and the IFN-β production was measured by ELISA assay (G, I). Data are shown as mean ± SD (*n* = 3 biological replicates), unpaired *t* test. (H) Data are shown as mean ± SD (*n* = 3), unpaired *t* test (G, I). Source data are available online for this figure.

## Deficiency of HDAC6 protects mice against HSV-1 infection

To further confirm the role of HDAC6 in regulating the antiviral immune response and efficient HSV-1 infection in vivo, we examined HSE pathogenesis in *Hdac6*−/− mice. We first analyzed the expression of HDAC6 in *Hdac6*−/− mice and found that HDAC6 expression was depleted in both trigeminal (TG) and brain tissue, accompanied by enhanced α-tubulin acetylation (Appendix Fig. S1A), as well as total acetylated proteins (Appendix Fig. S1B). Immunohistochemical analysis also revealed diminished expression of HDAC6 in the olfactory bulb (OB), cerebral cortex (CX), and medulla oblongata (P/M/C) of *Hdac6*−/− mice (Appendix Fig. S1C,D). To establish the HSE model, WT mice and *Hdac6*−/− mice were intranasally infected with HSV-1 (1 × 10⁷ PFU) (Fig. 3A), and the latter exhibited slighter brain and periocular swelling on day 5 of HSV-1 infection than WT mice (Fig. 3B). Additionally, the

weight loss observed in WT mice is more pronounced compared to *Hdac6*−/− mice (Fig. 3C). The RT-qPCR results showed a significant reduction in the viral DNA copy number in the TG and brain tissue of *Hdac6*−/− mice (Fig. 3D,E), as well as enhanced production of IFN-response genes (Appendix Fig. S1E), compared with WT mice. Moreover, H&E staining showed a marked decline cytoplasmic vacuolization in brain tissue following HSV-1 infection in HDAC6-deficient mice (Appendix Fig. S1F). Furthermore, *Hdac6*−/− mice exhibited heightened activation of the STING-TBK1-IRF3 axis in the OB, CX and P/M/C tissues following HSV-1 infection (Fig. 3F). These findings indicate that HDAC6 impedes antiviral innate immunity, thus promoting HSV-1 infection and HSE pathogenesis.

## HDAC6 interacts with and deacetylates TRIM56 at K110

Next, we explored the molecular mechanism through which HDAC6 negatively regulates the antiviral innate immune response

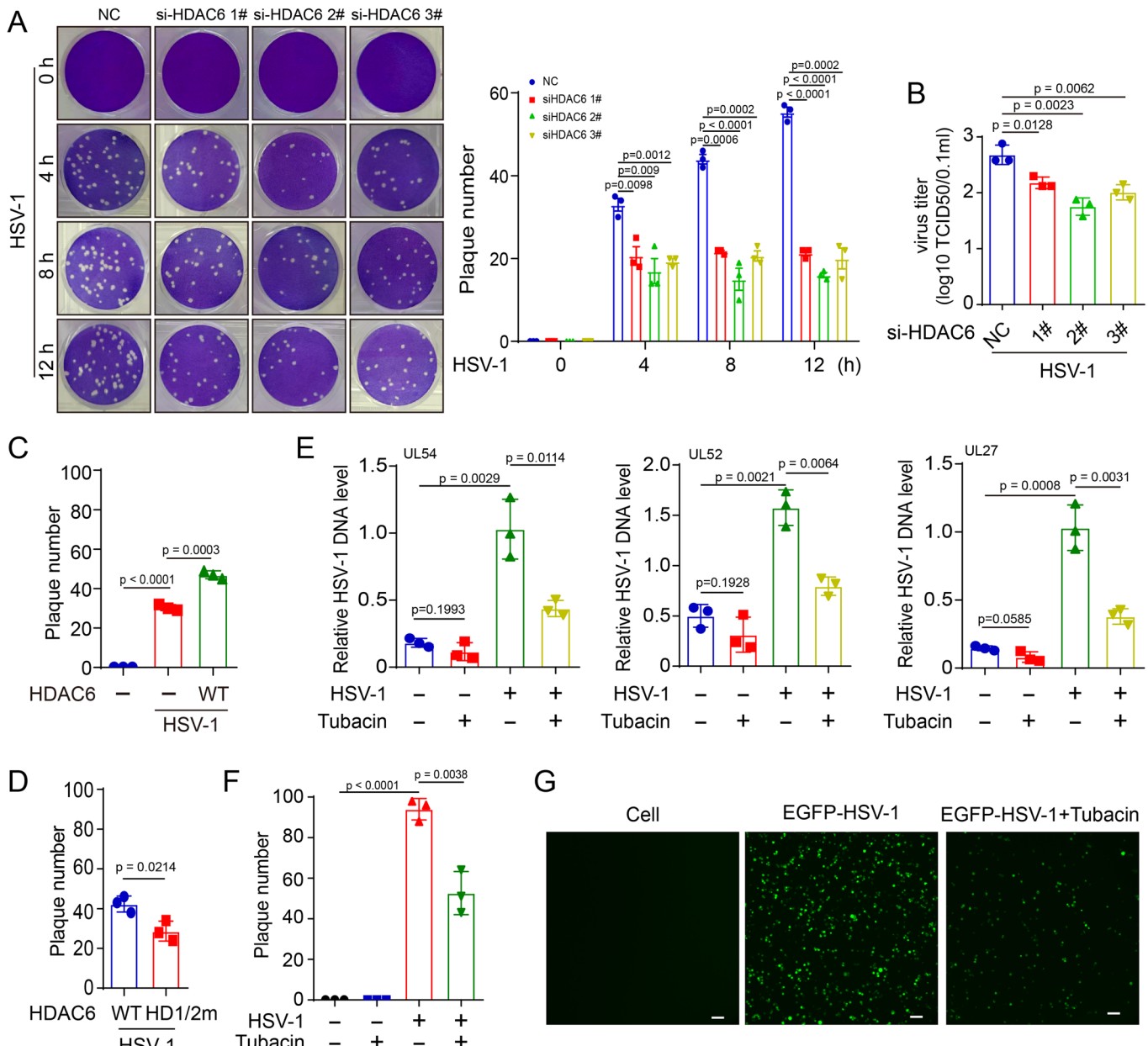

**Figure 2. HDAC6 promotes HSV-1 infection.**

(A, B) N2a cells were transfected with negative control (NC) siRNA or HDAC6 siRNA for 72 h followed by infection with HSV-1 (MOI = 3) for 12 h. The formation of viral plaques (A) and viral titers (B) were measured. Data were analyzed using the unpaired t test, which are shown as mean ± SD (n = 3 biological replicates). (C, D) N2a cells were transfected with vector, HDAC6 WT or HDAC6 deacetylase activity mutation (HD1/2 m) for 72 h followed by infection with HSV-1 (MOI = 3) for 12 h. The formation of viral plaques was measured. Data were analyzed using the unpaired t test, which are shown as mean ± SD (n = 3 biological replicates). (E, F) Effect of tubacin (1 μM) on the DNA copy number of viral genes (E) and viral plaque formation (F). Data were analyzed using unpaired t test, which are shown as mean ± SD (n = 3 biological replicates). (G) Effect of tubacin (1 μM) on EGFP-HSV-1 (MOI = 1) infection for 24 h, scale bars, 200 μm. Source data are available online for this figure.

by using proteomics. We performed HDAC6-based immunoprecipitation and LC–MS/MS (HDAC6 Interactome) to identify the interacting proteins of HDAC6 in N2a cells, and discovered 2,028 potential HDAC6 interactors (Fig. 4A; Dataset EV1). The reactome enrichment analysis revealed that these interacting proteins were primarily involved in various cellular processes, such as protein synthesis, virus infection and the cell cycle (Fig. 4B). Subsequently, we conducted acetyl-lysine immunoprecipitation and LC–MS/MS

(Acetylome) on N2a cells treated with tubacin for 2 h (Fig. 4C). This enabled the identification of 364 differentially regulated acetylated peptides from 252 proteins (fold change >1.2, 304 increased and 60 decreased in abundance) (Fig. 4D; Dataset EV2). It is noteworthy that 83 proteins from the Acetylome displaying altered abundance of acetyl-lysine have been identified as part of the HDAC6-Interactome. Seven of these acetylated proteins, including DDX3X, PTBD, TRIM56, IPO7, PPID, SHMT2, and

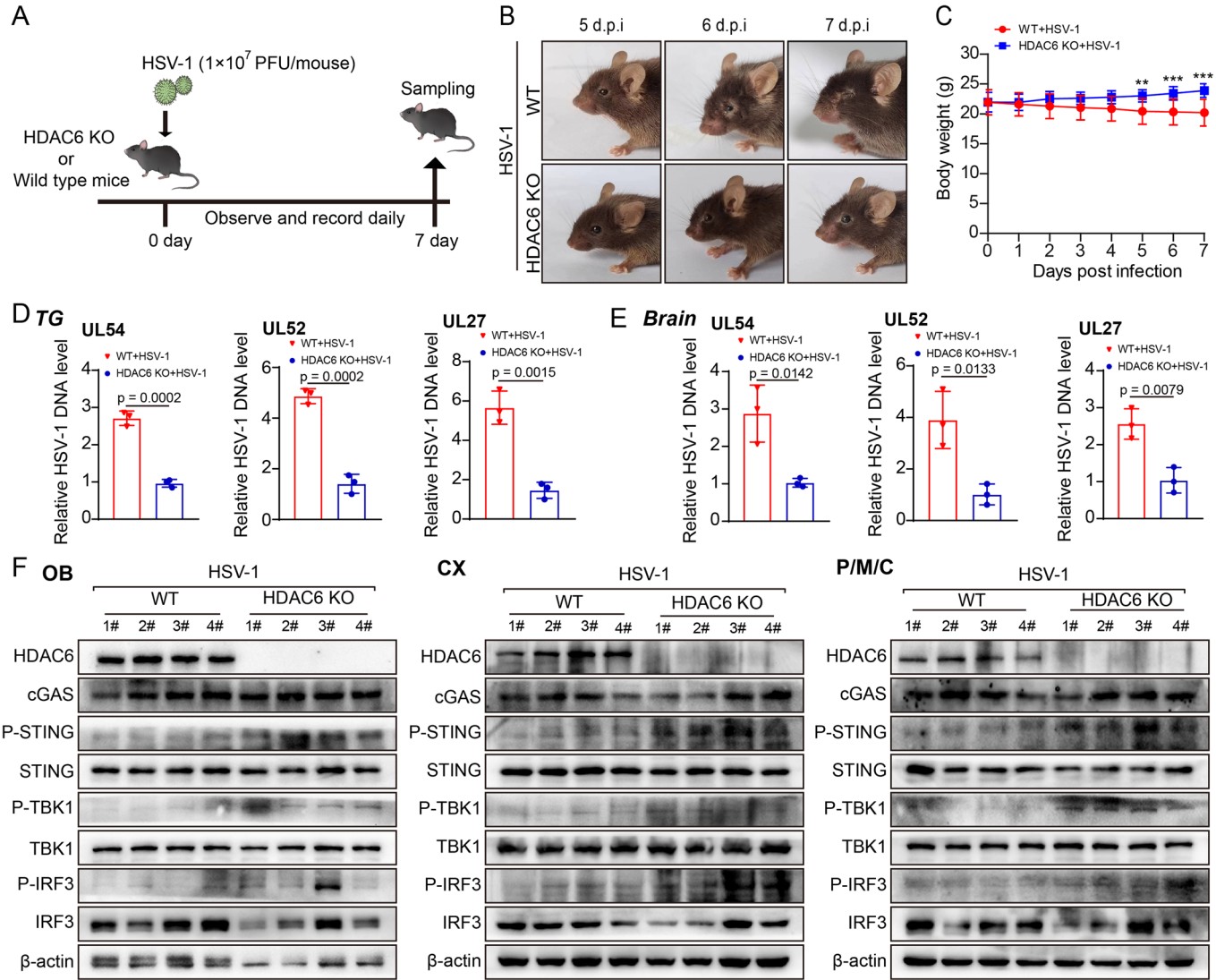

**Figure 3. Deficiency of HDAC6 protects mice against HSV-1 infection.**

(A) Schematic diagram showing the establishment of the HSE model ($n = 10$ mice per group). (B) Photographic records of the eyes and brain symptoms of mice ($n = 4$ mice per group). (C) Body weight of mice was recorded daily. Data were analyzed using the two-way ANOVA, which are shown as mean ± SD ($n = 10$ mice per group). **$P = 0.0073$ (day 5), ***$P = 0.0007$ (day 6), ***$P < 0.0001$ (day 7). (D) RT-qPCR analysis of the DNA copy number of viral genes in the trigeminus. Data were analyzed using the unpaired $t$ test, which are shown as mean ± SD ($n = 3$ mice per group). (E) RT-qPCR analysis of the DNA copy number of viral genes in the brain tissues. Data were analyzed using the unpaired $t$ test, which are shown as mean ± SD ($n = 3$ mice per group). (F) Western blot analysis of proteins involved in the cGAS-STING signaling in the OB, CX and PMC tissues ($n = 4$ mice per group). Source data are available online for this figure.

ZC3HAV1, have been implicated in innate immunity and viral infection, based on functional annotation using databases such as GO Biological Process, KEGG and reactome (Fig. 4E).

To assess the changes in acetylated proteins during early HSV-1 infection, we performed time-dependent acetylomics to quantify protein acetylation during the early stages of HSV-1 infection (0 h, 2 h, 4 h, 6 h) (Fig. EV3A). Through both correlation and principal component analyses (PCA), it was determined that the data correlation among various histological samples exceeded 0.94 (Appendix Fig. S2A) and that there was a notable distinction among the PCA analysis within each group (Fig. EV3B). The sample clustering analysis also did not exhibit any outliers (Appendix Fig. S2B). Then, weighted gene coexpression network analysis (WGCNA) was employed to obtain the correlation

patterns and coexpressed modules of acetylation modification sites between distinct time points (Dataset EV3). By utilizing the topological overlap matrix and hierarchical average chained clustering method, we were able to identify six modules by selecting the power of $\beta = 22$ (scale-free $R^2 = 0.7$) as the soft threshold to obtain an approximation of the scale-free topology (Appendix Fig. S2C). Each module is displayed in a distinct color, and the loci of each module exhibit comparable coexpression characteristics (Appendix Fig. S2D). We summarized the coexpression of the featured loci, based on which we calculated the correlation of each featured locus with different viral infection time points. We identified feature modules corresponding to different virus infection time points using the module-virus infection time point relationship graph (Fig. EV3C). For instance, at 6 h of infection, the

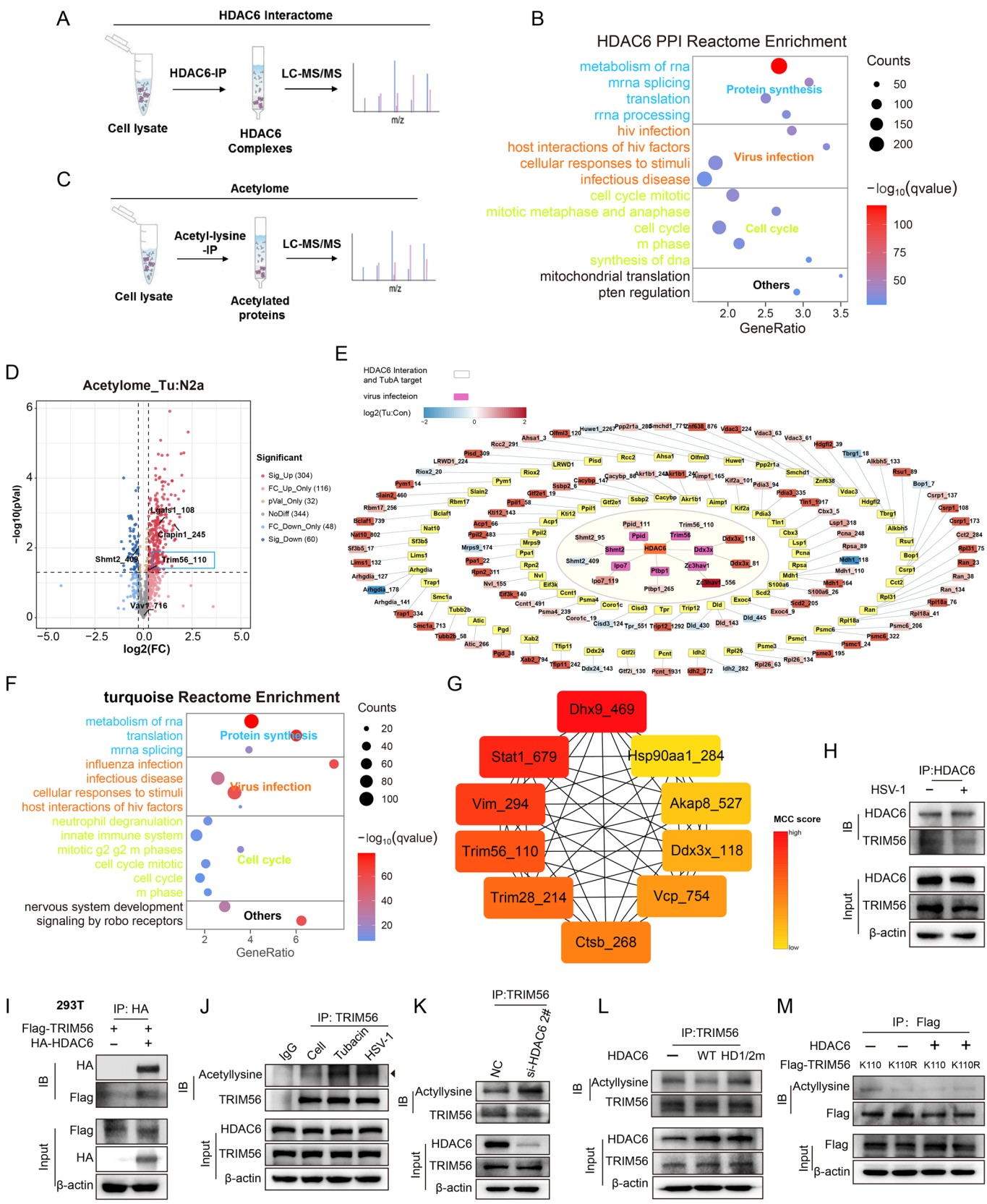

**Figure 4. HDAC6 interacts with and deacetylates TRIM56 at K110.**

(A) HDAC6-based immunoprecipitation and LC–MS/MS (HDAC6 Interactome) to identify the interacting proteins of HDAC6 in N2a cells. (B) The Reactome enrichment analysis of HDAC6-interacting proteins. (C) Acetyl-lysine immunoprecipitation and LC–MS/MS (Acetylome) on N2a cells treated with or without tubacin (1 μM) for 2 h. (D) Volcano plot displays the differential acetylation of amino acid sites, with red representing up-regulation and blue indicating downregulation (P < 0.05, log2 fold change ≥0.263). (E) Network diagram shows 83 HDAC6-inteacting proteins regulated by tubacin. Grey-bordered rectangles represent HDAC6-interacting proteins targeted by tubacin (TubA). The color changes in rectangles without grey borders signify the differences in acetylated site values following tubacin treatment. The purple rectangles depict antiviral-associated proteins that are targeted by tubacin and interact with HDAC6 proteins. (F) The Reactome enrichment analysis of acetylated proteins in the turquoise module. (G) The top 10 essential sites of the turquoise module were identified using the MCC algorithm. The importance of the sites in the network is indicated by the color saturation. (H) Co-IP analysis of the interaction between HDAC6 and TRIM56 in N2a cells infected with HSV-1 (MOI = 3) for 12 h. (I) Co-IP analysis of the interaction between HDAC6 and TRIM56 in 293T cells co-transfected with HA-HDAC6 and Flag-TRIM56. (J–L) Co-IP analysis of the acetylated TRIM56 in N2a cells treated with tubacin (1 μM) for 12 h (J), transfected with HDAC6 siRNA (K), transfected with HDAC6 WT or HD1/2 m (L). The black arrow indicates the protein band of acetylated TRIM56. (M) N2a cells were transfected with Flag-TRIM56 WT or K110R for 72 h followed by infection with HSV-1 (MOI = 3) for 12 h. Co-IP analysis of the acetylated Flag-TRIM56 in cell lysates. Source data are available online for this figure.

turquoise coexpression module exhibited a significant and positive correlation (correlation: 0.99, P < 0.0001) (Fig. EV3D). Our focus therefore lay on the acetylation modification sites present in this module. Analysis of the Reactome database found that the acetylated proteins in the turquoise module, which is the most relevant coexpression module during the 6-hour viral infection, are primarily associated with protein synthesis, virus infection, and cell cycle-related pathways (Fig. 4F). We employed Cytoscape software to construct a network of acetylation sites that were related to innate immunity and viral infection within the turquoise module, and the top 10 hub sites in the coexpression network were identified using cytoHubba according to MCC scores after 6 h of viral infection (Fig. 4G). Significantly, only K110 of TRIM56 and K118 of Ddx3x were identified in the HDAC6 Interactome and Acetylome. As TRIM56 has been demonstrated to ubiquitinate cGAS and STING to promote type I IFN production, we further explored whether deacetylation of TRIM56 at K110 by HDAC6 could affect its role in the intracellular DNA-mediated innate immune response.

By co-immunoprecipitation assay, we found that HSV-1 infection reduced the interaction between HDAC6 and TRIM56 in N2a cells (Fig. 4H), and the interaction was confirmed in 293 T cells co-transfected with HA-HDAC6 and Flag-TRIM56 (Fig. 4I). In addition, tubacin treatment and HDAC6 knockdown significantly increased the acetylation level of TRIM56 in N2a cells (Fig. 4J,K), and in BV2 cells (Fig. EV3E,F). Furthermore, overexpression of HDAC6 WT decreased TRIM56 acetylation, whereas HDAC6-HD1/2 m upregulated the acetylation of TRIM56 (Fig. 4L). Notably, upon HSV-1 infection, the acetylation of TRIM56 was significantly upregulated (Fig. EV3G). To further investigate the potential regulatory role of HDAC6 in the acetylation of TRIM56 K110, we conducted co-transfection experiments with HDAC6 and either Flag-TRIM56 K110 or Flag-TRIM56 K110R (non-acetylated variant of TRIM56 K110). Immunoprecipitation analysis revealed an increase in the acetylation level of Flag-TRIM56 K110 during HSV-1 infection compared to Flag-TRIM56 K110R, which was subsequently decreased following co-transfection with HDAC6 (Fig. 4M). Thus implicating the direct regulation of TRIM56 deacetylation by HDAC6 during HSV-1 infection.

## The acetylation of TRIM56 K110 is critical for its binding to HDAC6 and regulation of the IFN response

To confirm the role of TRIM56 in the antiviral response, we used TRIM56 siRNAs to suppress the expression of TRIM56 and found that downregulation of TRIM56 significantly inhibited TBK1 and

IRF3 activation triggered by HSV-1 infection (Appendix Fig. S3A), and effectively decreased the expression of Ifnb1 stimulated by CT-DNA and cGAMP (Appendix Fig. S3B). Consistently, TRIM56 knockdown inhibited IFN-β production unpon CT-DNA, ISD and cGAMP stimulation (Appendix Fig. S3C). In addition, TRIM56 knockdown inhibited IFN-β production in a time-dependent manner (Appendix Fig. S3D), and promoted viral infection (Appendix Fig. S3E). Therefore, these results further supported that TRIM56 is a positive IFN regulator. The K110 residue of TRIM56 is located in the B box-type 1 domain (TRIM56 B-box1) (Fig. 5A), and whether this domain plays a significant role in the catalytic function of TRIM56 in antiviral regulation remains unclear. Notably, we found that overexpression of TRIM56 B-box1 domain apparently enhanced HSV-1-induced IFN-β production, whereas overexpression of the HDAC6-CD1/2 (the deacetylase domain) reduced IFN-β production (Fig. 5B). Besides, co-transfection of HDAC6-CD1/2 and TRIM56 B-box1 resulted in significant inhibition of IFN-β (Fig. 5B), similar to the effect of full-length HDAC6, implying that HDAC6 deacetylase activity inhibits TRIM56 B-box1 function. Therefore, these results further supported the hypothesis that HDAC6 modulates the TRIM56-mediated antiviral response. In addition, antisense-oligonucleotide (ASO) was employed to downregulate the expression of TRIM56 B-box1, which effectively reduced the gene expression of TRIM56 (Fig. 5C), decreased TBK1 and IRF3 activation triggered by HSV-1 infection (Fig. 5D) and reduced the production of IFN-β (Fig. 5E). Moreover, the exogenous overexpression of B-box1 was observed to markedly enhance IFN-β production, whereas FL-TRIM56 demonstrated a more pronounced capacity to elevate IFN-I production (Fig. 5F). To ascertain the potential for TRIM56-mediated monoubiquitination of cGAS, N2a cells were transfected with FL-TRIM56 or TRIM56 B-box1, subjected to immunoprecipitation with anti-cGAS antibody, and then analysed by immunoblotting with a monoubiquitin-specific VU-1 antibody (Seo et al, 2018). A fraction of cGAS was readily detected by VU-1 antibody and this VU-1 antibody reactivity was further enhanced by FL-TRIM56 expression in comparison to B-box1. (Fig. 5G). Therefore, these results suggest that B-box1 is critical for TRIM56-mediated regulation of the IFN response, which is further inhibited by HDAC6 deacetylase activity. To assess the role of K110 deacetylation in the antiviral response, we constructed a K110A B-box1 mutant (lysine to alanine substitution), which is incapable of undergoing acetylation. It is worth noting that the K110A mutant, in contrast to TRIM56 B-box1 WT,

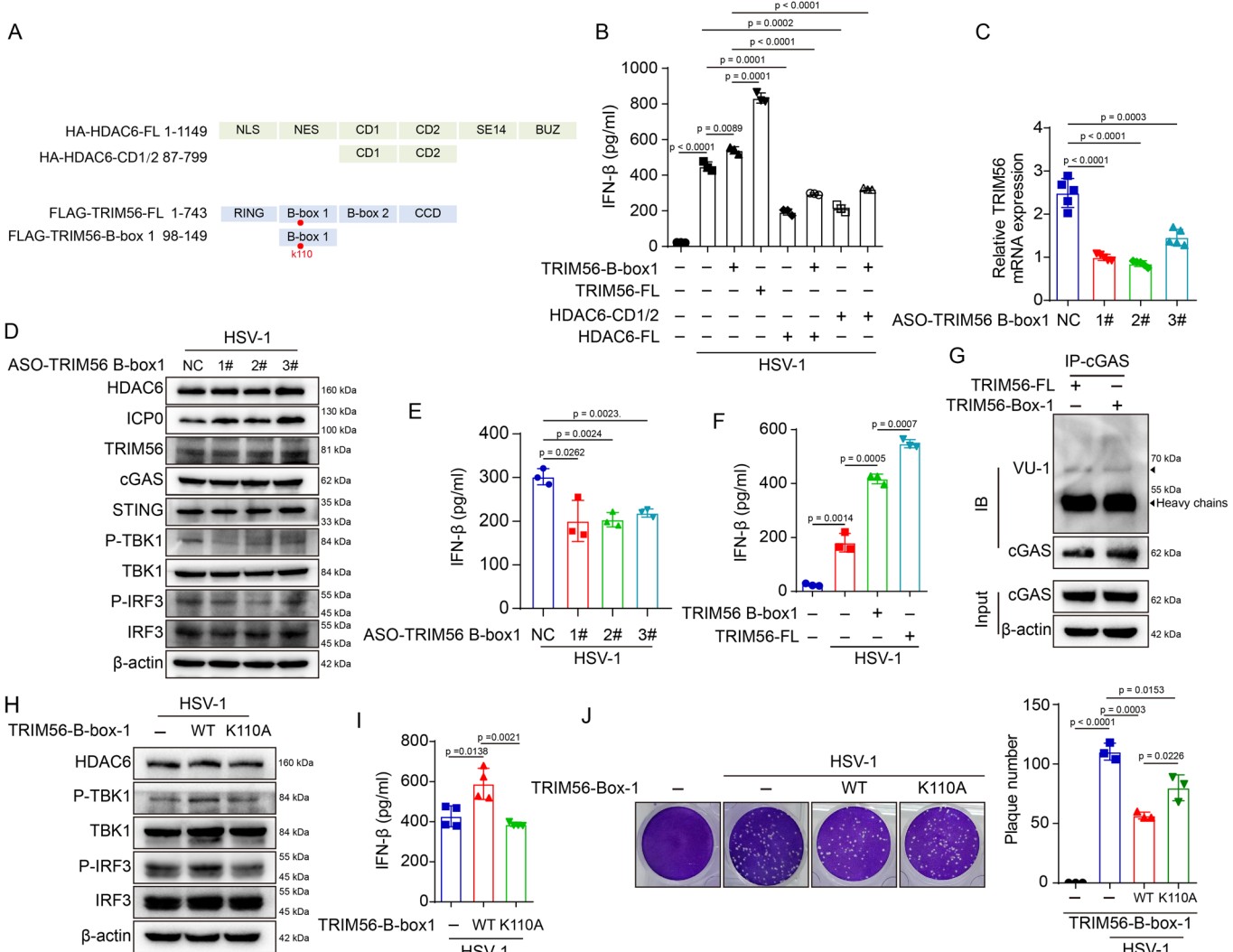

**Figure 5. Acetylation of K110 within the TRIM56 B-box1 domain interacts with the HDAC6 and regulates IFN-β production.**

(A) Schematic illustration of HDAC6 and TRIM56 structure. FL, full length; CD1/2, histone deacetylase 1 and histone deacetylase 2 domain; B-box1, B box-type 1 domain. (B) N2a cells were infected with HSV-1 (MOI = 3) for 12 h after overexpression of HDAC6-FL, TRIM56-FL, HDAC6-CD1/2 or TRIM56 B-box1, and ELISA assay was conducted to detect IFN-β production. Data were analyzed using the unpaired t-test, which are shown as mean ± SD (n = 3 biological replicates). (C–E), N2a cells were transfected with ASO-TRIM56 B-box1. (C) RT-qPCR assay of TRIM56 mRNA expression. Data were analyzed using the unpaired t-test, which are shown as mean ± SD (n = 5 biological replicates). (D) Western blotting was used to assess the levels of indicated proteins in N2a cells infected with HSV-1 (MOI = 3) for 12 h. (E) ELISA was used to assess the levels of IFN-β in the culture supernatants. Data were analyzed using the unpaired t test, which are shown as mean ± SD (n = 3 biological replicates). (F, G), N2a cells transfected with TRIM56-FL or TRIM56 B-box1 for 72 h followed by infection with HSV-1 (MOI = 3) for 12 h. (F) ELISA was used to assess the levels of IFN-β in the culture supernatants. Data were analyzed using the unpaired t test, which are shown as mean ± SD (n = 3 biological replicates). (G) Co-IP assay of cGAS monoubiquitination. The black arrow indicates the protein band of cGAS monoubiquitination. (H–J) N2a cells were infected with HSV-1 (MOI = 3) for 12 h after overexpression of TRIM56 WT or K110A, and the effects on the cGAS-STING signaling pathway (H), IFN-β production (I), and viral plaque formation (J) were analyzed. Data were analyzed using the unpaired t test, which are shown as mean ± SD (n = 3 biological replicates). Source data are available online for this figure.

impeded the TBK1 and IRF3 activation that resulted from HSV-1 infection (Fig. 5H), decreased IFN-β production (Fig. 5I) and simultaneously promoted viral plaque formation (Fig. 5J). Therefore, these findings suggest that B-box1 K110 acetylation is critical for the antiviral response.

To further investigate the impact of K110 acetylation on the interaction between HDAC6 and TRIM56, we conducted molecular docking and molecular dynamic simulations. As shown in Fig. EV4A, the B-box1 domain of TRIM56 (grey) is tightly interacts with the CD1-CD2 structural domain of HDAC6 (cyan).

Subsequently, via alanine scanning at the interaction interface of HDAC6 and TRIM56, we observed that the substitution of K110 with A110 had a considerable impact on the interaction between HDAC6 and TRIM56 (Fig. EV4B). The interactive complexes underwent a 100 ns molecular dynamics simulation to ascertain the stability of protein conformational dynamics and folding compactness by evaluating alterations in the root mean square deviation (RMSD) and radius of gyration (Rg). The RMSD curve demonstrated a significantly lower fluctuation in the HDAC6/TRIM56 WT complex compared to the HDAC6/TRIM56 K110A

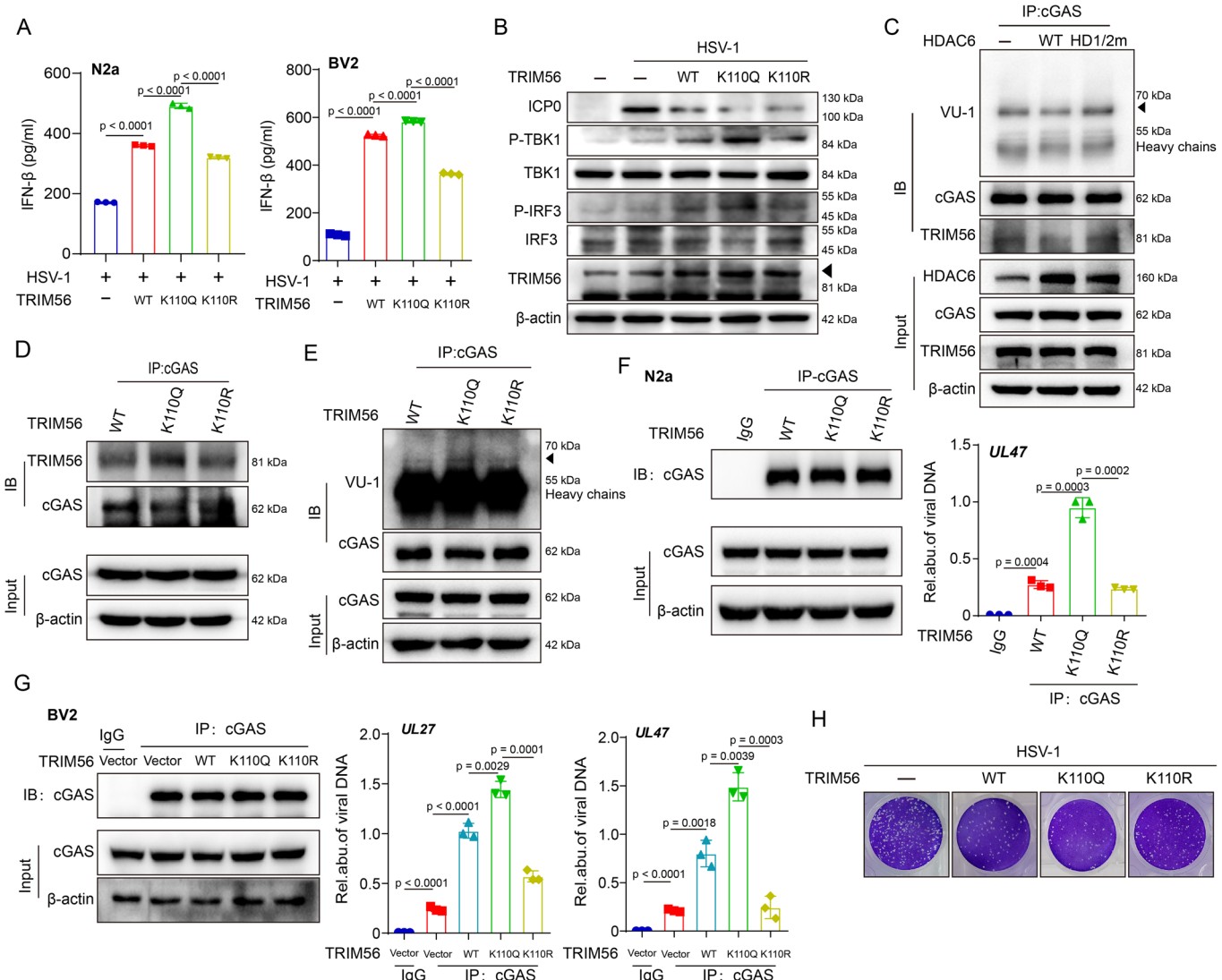

**Figure 6. HDAC6 deacetylates TRIM56 at K110 to inhibit innate immune responses.**

(A) ELISA analysis of IFN-β production in cells that were transfected withTRIM56 WT, K110Q or TRIM56 K110R followed by infection with HSV-1 (MOI = 3) for 12 h. Data were analyzed using the unpaired $t$ test, which are shown as mean ± SD ($n = 3$ biological replicates). (B) Western blot analysis of TBK1-IRF3 in N2a cells transfected with TRIM56 WT, K110Q or TRIM56 K110R followed by infection with HSV-1 (MOI = 3) for 12 h. The black arrow indicates the protein band of TRIM56. (C) Co-IP analysis of TRIM56 and cGAS in cells transfected with HDAC6-WT or HDAC6-HD1/2 m. The black arrow indicates the protein band of cGAS monoubiquitination. (D, E) Co-IP assay of the interaction between TRIM56 and cGAS (D) and cGAS monoubiquitination (E) in cells transfected with TRIM56 WT, K110Q or K110R followed by infection with HSV-1 (MOI = 3) for 12 h. The black arrow indicates the protein band of cGAS monoubiquitination. (F, G) Co-IP and RT-qPCR analysis of cGAS-bound viral DNA (UL47 or UL27) in N2a cells (F) or BV2 cells (G) transfected with TRIM56 WT, K110Q or K110R followed by infection with HSV-1 (MOI = 3) for 12 h. Data were analyzed using the unpaired $t$ test, which are shown as mean ± SD ($n = 3$ biological replicates). (H) Viral plaque assays in N2a cells transfected with TRIM56 WT, K110Q or K110R followed by infection with HSV-1 (MOI = 3) for 12 h. Source data are available online for this figure.

counterparts (Fig. EV4C), implying a more stable conformation. Similarly, the Rg curve exhibited a corresponding stable conformation of the HDAC6/TRIM56 WT in contrast to the mutant (Fig. EV4D), indicating that K-to-A substitution reduces the interaction between HDAC6 and TRIM56. Furthermore, the calculation of binding free energy utilizing MM-GBSA presents a reliable framework for verifying the affinity between the ligand and receptor, especially in the context of protein-protein interactions (Genheden and Ryde, 2015). The calculated value of the binding free energy (Δ total) for WT (−50.38 kcal/mol) is greater compared

to K110A (−29.76 kcal/mol), indicating that WT has a more stable binding affinity against HDAC6 when compared to K110A (Appendix Table S1). In addition, the van der Waals (VDWAALS) and electrostatic energy levels (EEL) make significant contributions to the complex binding. Notably, the Van der Waals energy ranged from −85.59 kcal/mol for WT-HDAC6 to −67.42 kcal/mol for K110A-HDAC6, while the electrostatic energy ranged from −900.11 kcal/mol for WT-HDAC6 to −424.04 kcal/mol for K110A-HDAC6. Therefore, HDAC6 has more extensive chemical interactions and stability at the docked site, showing a notably

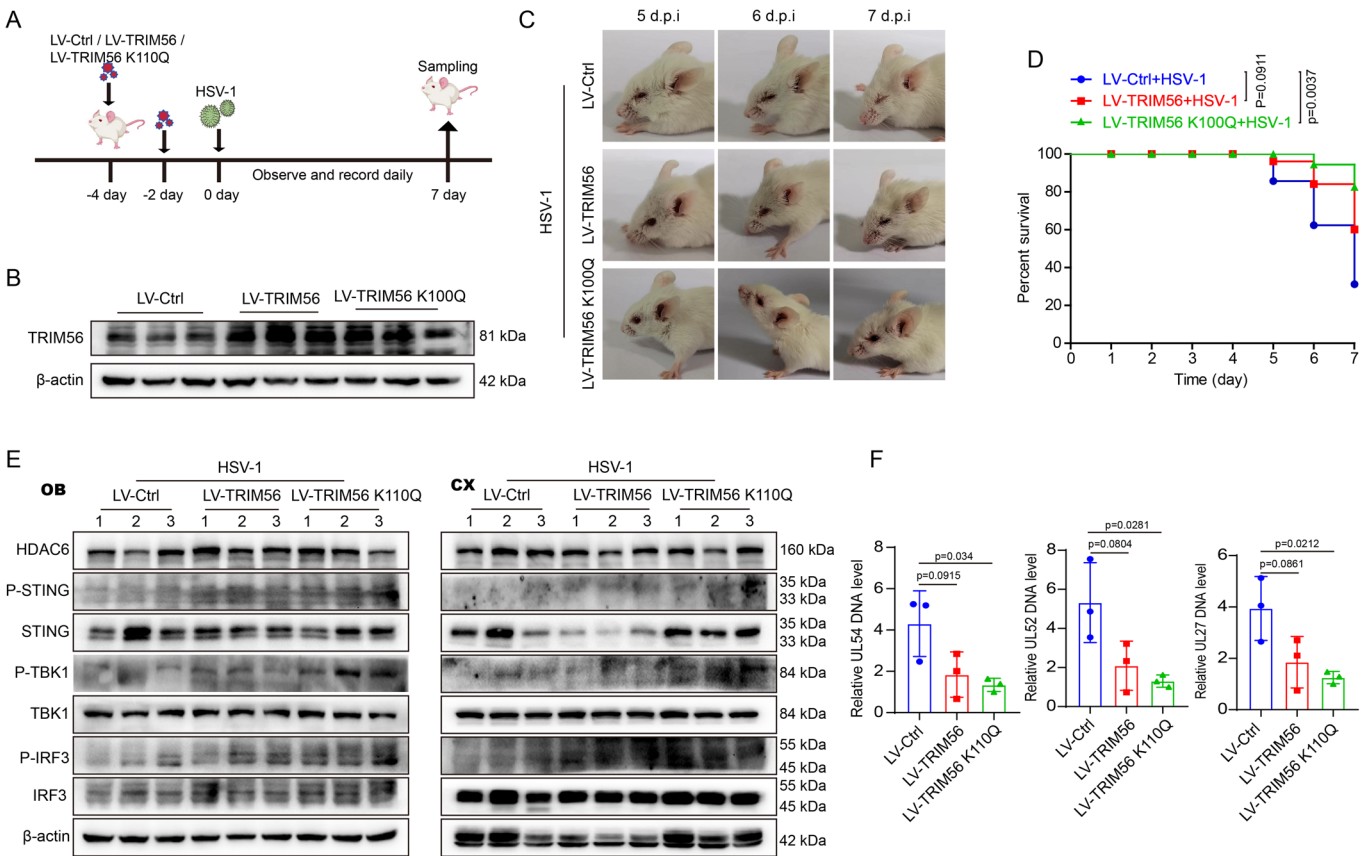

**Figure 7. Enhanced acetylation of TRIM56 K110 activates innate immune response against HSV-1 infection in vivo more efficiently.**

(A) Schematic diagram illustrating the HSE model in mice that overexpress TRIM56 ($n = 10$ mice per group). (B) Western blot assay measures TRIM56 levels in mouse brain tissue 4 days after lentivirus infection ($n = 3$ mice per group). (C) Photographic records of eyes and brain symptoms from HSE mice ($n = 4$ mice per group). (D) Mouse survival rate were daily recorded. Data were analyzed using log-rank (Mantel-Cox) test, which are shown as mean ± SD ($n = 10$ mice per group). (E) Western blot analysis of the cGAS-STING signaling in OB and CX tissues of mice ($n = 3$ mice per group). (F) RT-qPCR analysis of the DNA copy number of viral genes *UL52*, *UL52* and *UL27* in brain tissues ($n = 3$ mice per group). Data were analyzed using the unpaired $t$ test, which are shown as mean ± SD ($n = 3$ mice per group). Source data are available online for this figure.

greater affinity for TRIM56 WT. In conclusion, the findings indicate that the acetylation/deacetylation status of TRIM56 K110 within the B-box1 has an impact on the interactive affinity of TRIM56 to HDAC6 and its regulatory effects on the antiviral IFN response.

## HDAC6 deacetylates TRIM56 K110 to attenuate the cGAS-mediated antiviral response

To further assess the effect of K110 acetylation on the TRIM56-mediated type I IFN response, we generated TRIM56 acetyl-mimetic mutants that were either acetylated (K110Q) or non-acetylated (K110R). Our results showed that the K110Q mutant upregulated IFN-β production induced by the virus when compared with TRIM56 WT (Fig. 6A). On the other hand, the K110R mutant decreased IFN-β production (Fig. 6A). Furthermore, the K110Q mutant enhanced the activation of TBK1 and IRF3 (Fig. 6B). These findings indicate that K110 acetylation plays a critical role in the impact of TRIM56 on IFN production. As

TRIM56 has been shown to monoubiquitinate cGAS, stimulating its dimerization, DNA binding and the consequent activation of the STING-mediated antiviral IFN response (Seo et al, 2018), we investigated whether HDAC6 deacetylates TRIM56, affecting cGAS monoubiquitination and the resulting IFN production. As shown in immunoblotting with a monoubiquitin-specific VU-1 antibody, the overexpression of HDAC6 WT suppressed TRIM56 binding to cGAS and cGAS monoubiquitination, whereas HDAC6-HD1/2 m augmented cGAS monoubiquitination (Fig. 6C). Furthermore, the K110Q mutant intensified the interaction between TRIM56 and cGAS (Fig. 6D), leading to increased cGAS monoubiquitination (Fig. 6E), whereas the K110R mutant exhibited no significant effect. Compared to TRIM56 WT, the K110Q mutant considerably enhanced cGAS binding to viral DNA, while the K110R mutant markedly decreased TRIM56's binding of cGAS to viral DNA in N2a cells and BV2 cells compared to the K110Q mutant (Fig. 6F,G). In addition, the K110Q mutant notably hindered the formation of viral plaques (Fig. 6H). Together, these findings indicate that deacetylation of TRIM56 K110 by HDAC6 hinders the cGAS-mediated antiviral innate immune response.

## Species-specific amino acids at position 110 of TRIM56 differentially regulate the IFN response

Interestingly, sequence comparison analyses revealed a discrepancy in the amino acid residues present at position 110 of TRIM56 in human and mouse. Specifically, human TRIM56 (hTRIM56) has threonine (T), whereas mouse TRIM56 (mTRIM56) has lysine (K) (Fig. EV5A). Molecular docking analysis revealed that either TRIM56 K110 (mTRIM56) or TRIM56 T110 (hTRIM56) interacted directly with cGAS (Fig. EV5B), indicating that the K-to-T substitution did not directly affect the TRIM56-cGAS interaction. However, the RMSD curve indicated a general decrease in conformational fluctuations within the cGAS/TRIM56 T110 complexes compared to the cGAS/ TRIM56 K110 counterparts (Fig. EV5C). Additionally, the Rg curve showed no significant difference between the two variants (Fig. EV5D). Nevertheless, the TRIM56 T110 displayed considerably more RMSD fluctuations in the first 60 ns. These significant variations are able to reinforce the induction-fit phenomenon, leading to improved conformations for cGAS/TRIM56 interactions and ultimately enhancing the stability of the cGAS/TRIM56 complexes. Furthermore, the mouse variant exhibited a binding free energy value ($-30.98$ kcal/mol) greater than that of the human variant ($-40.83$ kcal/mol) (Appendix Table S2). Additionally, the human variant had better VDWAALS and EEL interactions with cGAS compared to the WT. Thus, these findings suggest that the TRIM56 T110 displays significantly increased cGAS affinity and consequently hTRIM56 triggers a more effective cGAS-mediated antiviral response than mTRIM56. Finally, we produced a B-box1 K110 mutant (B-box1 K110T) to mimic the hTRIM56 in mice and observed a more substantial enhancement of TBK1 and IRF3 activation (Fig. EV5E) and a significant decrease in viral plaque formation (Fig. EV5F). In addition, we overexpressed TRIM56 in hTRIM56 deleted HMC3 cells and found that hTRIM56 promoted IFN-I production compared with mTRIM56 during infection with HSV-1 (Fig. EV5G). Together, these findings imply that humans are more resistant to HSV-1 infection and that TRIM56 K110 potentially serves as a critical node for species-specific cGAS-mediated antiviral innate immune activation.

## TRIM56 K110 acetylation regulates efficient antiviral immune response in vivo

To further confirm the role of TRIM56 K110 deacetylation in regulating the antiviral immune response in vivo, we established the HSE model in mice overexpressing TRIM56 WT or K110Q as previously described (Wang et al, 2022). Lentivirus (puro)-CMV-NC, lentivirus (puro)-CMV-TRIM56 and lentivirus (puro)-CMV-TRIM56 k110Q were intranasally administered twice (with a 48-hour interval) prior to the HSE model, respectively (Fig. 7A). On the fourth day of lentivirus infection, the mice exhibited normal symptoms (Appendix Fig. S4A) and their body weight continued to rise steadily (Appendix Fig. S4B). Furthermore, histological examination via H&E staining demonstrated that intranasal lentivirus infection did not cause any harm to the mouse heart, liver, spleen, lung, or kidney tissues (Appendix Fig. S4C). Western blot analysis demonstrated a significant upregulation of TRIM56 protein expression in the brain tissues of overexpressed mice (Fig. 7B). After lentivirus infection, the mice were intranasally

infected with HSV-1 to establish the HSE model. As expected, overexpression of TRIM56 WT notably reduced severe disease development, while overexpression of TRIM56 K110Q had a more significant impact on inhibiting hydrocephalus than that of TRIM56 WT in HSE mouse (Fig. 7C). In addition, compared with LV-TRIM56 WT or LV-Ctrl mice, a lower mortality rate and a slower weight loss were observed in LV-TRIM56 K110Q mice in HSE mice (Fig. 7D; Appendix Fig. S4D). Furthermore, the outcomes for hydrocephalus (Appendix Fig. S4E), hair loss (Appendix Fig. S4F), and the disease score of neurological symptoms (Appendix Fig. S4G) demonstrated that the LV-TRIM56 K110Q mice could effectively restrain HSV-1 infection. Additionally, the overexpression of TRIM56 K110Q enhanced TBK1 and IRF3 activation more than that of TRIM56 WT in mouse brain OB and CX tissues in HSE mouse (Fig. 7E). Consistent with the in vitro experiments, overexpressing TRIM56 K110Q further amplified the inhibitory effect of TRIM56 on viral DNA copy number (Fig. 7F). These findings suggest that K110 acetylation in TRIM56 is a crucial site for regulating the cGAS-mediated antiviral innate immune response.

## HSV-1 US3 phosphorylates HDAC6 to inhibit the cGAS-mediated antiviral response

To determine whether HSV-1 neurotropic infection affects HDAC6 deacetylase activity, we analyzed the protein levels of acetylated proteins by western blot assay. The results showed a decrease in the level of acetylated proteins without affecting total protein levels (ponceau staining) during the early stage of HSV-1 infection (Appendix Fig. S5A). In addition, a fluorometric kit was employed to assess the deacetylase activity of HDAC6, which showed a reduced AFC fluorescence intensity by HSV-1 early infection, whereas tubacin notably upregulated AFC fluorescence intensity (Fig. 8A). Meanwhile, HDAC6 expression was not significantly altered during viral infection in vitro (Appendix Fig. S5B–D), and remained constant in the HSE mouse model (Appendix Fig. S3E,F). These findings indicate that HSV-1 early infection enhances the deacetylase activity of HDAC6 to attenuate the cGAS-STING-mediated IFN response, thereby facilitating its infection.

Finally, we explored whether HSV-1 regulates the interaction between HDAC6 and TRIM56, and then affects the negative regulation of the antiviral immune response by HDAC6. The immunofluorescence assay showed that HDAC6 aggregates are present in the perinuclear area at 4 h post-infection (Fig. 8B). In addition, the nuclear HDAC6 was reduced, along with its cytoplasmic accumulation at 4 h post-infection (Fig. 8C). Importantly, the interaction between HDAC6 and TRIM56 was significantly upregulated at 4 h p.i. (Fig. 8D), suggesting that HSV-1 early infection promotes the nuclear-to-cytoplasmic transport of HDAC6 and enhances its interaction with TRIM56, thereby reducing TRIM56-mediated cGAS-STING activation. Consistently, the HDAC6 ΔNLS mutant, which is deficient in nuclear localization signal and is unable to enter the nucleus, inhibited TBK1 and IRF3 activation compared to HDAC6 WT during HSV-1 infection (Fig. 8E). In addition, the HDAC6 ΔNLS mutant apparently reduced the binding of cGAS to viral DNA (Fig. 8F) and promoted viral infection (Fig. 8G). Therefore, these results clearly demonstrated that the cytoplasmic localization of HDAC6 by HSV-1 impairs TRIM56-cGAS-mediated IFN response.

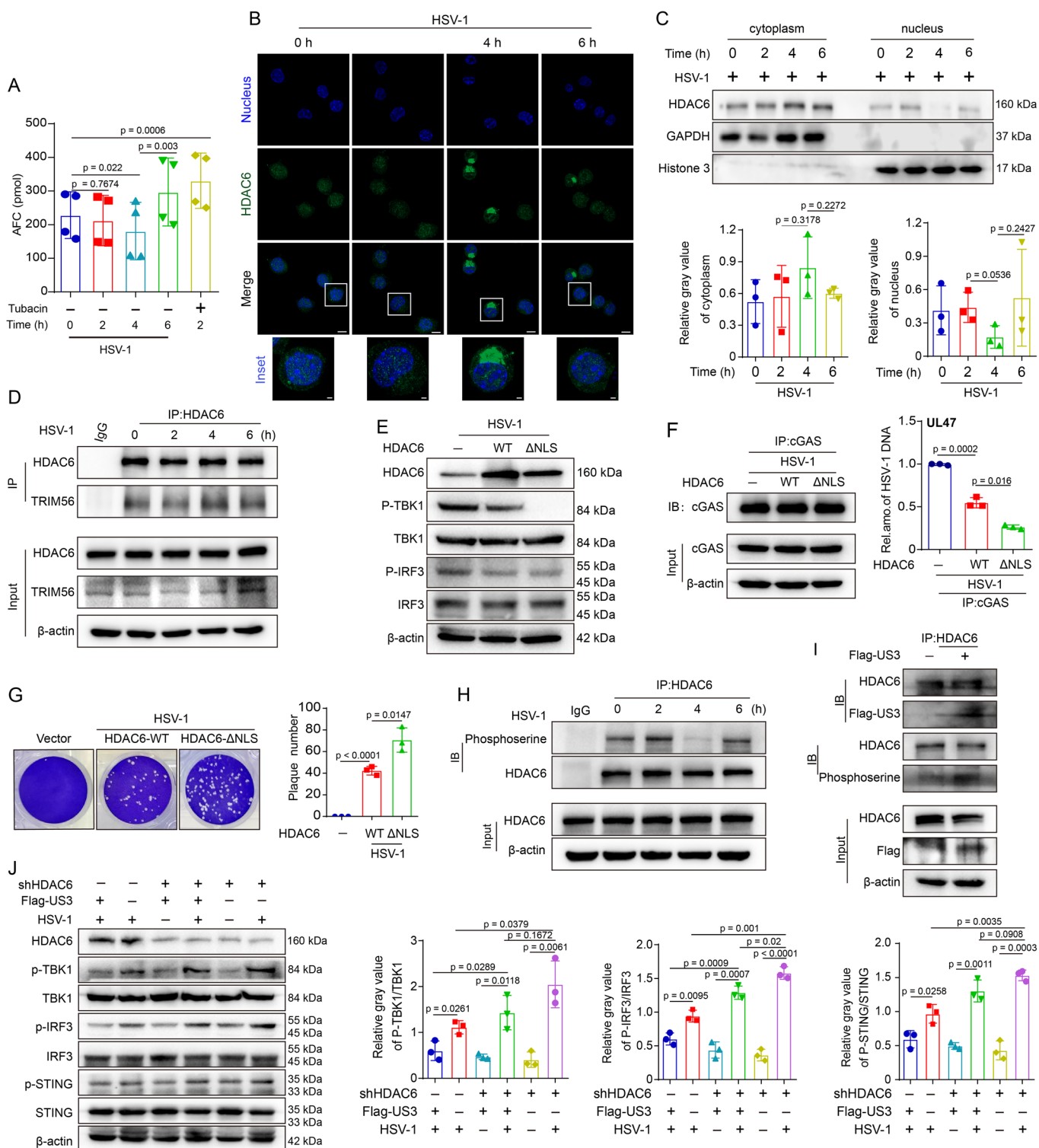

It has been reported that phosphorylation enhances HDAC6 deacetylase activity (Zheng et al, 2017). Interestingly, in the early stages of HSV-1 infection, we found that the phosphorylation level of HDAC6 was significantly enhanced (Fig. 8H). The viral protein US3 phosphokinase has been demonstrated to exert a negative regulatory effect on IFN production by phosphorylating RIG-I and

IRF3, which in turn suppress IFN production (Wang et al, 2013). In the HDAC6-interactome and acetylome, we also found that HDAC6 significantly interacted with US3 during early infection (Appendix Fig. S6A,B). Co-immunoprecipitation assay also showed that US3 interacted with HDAC6 and enhanced the serine phosphorylation levels of HDAC6 (Fig. 8I). In addition, exogenous

**Figure 8. HSV-1 US3 phosphorylates HDAC6 to enhance its deacetylase activity and cytosolic interactions with TRIM56.**

(A) HDAC6 enzymatic activity assay was conducted by measuring AFC fluorescence intensity following infection with HSV-1 (MOI = 5) or treatment with tubacin (1 µM) in indicated time. Data were analyzed using the unpaired $t$ test, which are shown as mean ± SD ($n = 4$ biological replicates). (B) Representative immunofluorescent images of HDAC6 (green) and nucleus (blue) in N2a cells infected with HSV-1 (MOI = 5) for the indicated time. Scale bars, 10 µm. The enlarged insets show the localization of HDAC6. Scale bars, 2 µm. (C) Western blot analysis of HDAC6 in cytosol and nucleus in N2a cells infected with HSV-1 (MOI = 5) for the indicated time, and histograms show quantified data (below). Data were analyzed using the unpaired $t$ test, which are shown as mean ± SD ($n = 3$ biological replicates). (D) Co-IP analysis the TRIM56-HDAC6 interaction in cells infected with HSV-1 (MOI = 5) for the indicated times. (E) Western blot analysis of IRF3 and TBK1 in N2a cells transfected with HDAC6 WT or ΔNLS for 72 h followed by infection with HSV-1 (MOI = 3) for 12 h. (F) Co-IP and RT-qPCR assay were performed to detect the cGAS-bound viral DNA in N2a cells transfected with HDAC6 WT or ΔNLS for 72 h followed by infection with HSV-1 (MOI = 3) for 12 h. Data were analyzed using the unpaired t-test, which are shown as mean ± SD ($n = 3$ biological replicates). (G) Viral plaque assays in N2a cells transfected with HDAC6 WT or ΔNLS for 72 h followed by infection with HSV-1 (MOI = 3) for 12 h. Data were analyzed using the unpaired $t$ test, which are shown as mean ± SD ($n = 3$ biological replicates). (H) HDAC6 and its phosphoserine were analyzed through Co-IP assay upon HSV-1 (MOI = 10) infection at different times. (I) Co-IP assay analysis of the US3-HDAC6 interaction and phosphoserine in N2a cells transfected with Flag-US3 for 48 h. (J) N2a cells were transfected with shRNA HDAC6 for 72 h and later transfected with Flag-US3 for 48 h. Western blotting was used to assess the levels of indicated proteins in N2a cells infected with HSV-1 (MOI = 3) for 12 h. Histograms show the quantification of P-TBK1 or P-IRF3 levels relative to that of TBK1 or IRF3. Data were analyzed using the unpaired $t$ test, which are shown as mean ± SD ($n = 3$ biological replicates). Source data are available online for this figure.

overexpression of US3 restrained TBK1 and IRF3 activation during HSV-1 infection. However, the inhibitory effect of US3 was attenuated by the knockdown of HDAC6 (Fig. 8J), indicating that US3 exerts its negative effect on IFN-I production, at least in part, through HDAC6. Furthermore, overexpression US3 alone did not significantly promote the activation of TBK1 and IRF3 (Appendix Fig. S6C), consistent with previous studies that US3 hyperphosphorylates IRF3 and RIG-I to maintain them in a signaling-repressed state (van Gent et al, 2022; Wang et al, 2013). Therefore, the above results suggest that US3 interacts with and phosphorylates HDAC6, thereby regulating the TRIM56-mediated antiviral immune response.

## Discussion

cGAS is a multifunctional cytoplasmic DNA sensor that binds endogenous or pathogen-derived dsDNA and induces the expression of IFN, a key event in regulating cellular senescence, autophagy, cancer development and viral defense (Gui et al, 2019; Li and Chen, 2018; Zheng et al, 2020). Viruses have evolved various strategies to interrupt the cGAS-STING pathway, inhibiting the IFN-mediated antiviral response and evading immune surveillance (Anwar et al, 2021; Jin et al, 2023). In this study, we identified HDAC6 as a novel regulator of the cGAS-STING pathway via TRIM56 deacetylation. Inhibition of HDAC6's deacetylase activity or HDAC6 deletion markedly promoted the activation of the cGAS-STING axis, resulting in increased IFN production and decreased HSV-1 infection. In addition, *Hdac6* KO mice were more resistant to HSV-1 infection and HSE pathogenesis. Our data clearly demonstrated that HDAC6 interacts with TRIM56 at K110 and deacetylates it, resulting in reduced monoubiquitination and DNA binding of cGAS. The acetylation of TRIM56 at K110 significantly affected its binding to HDAC6 and regulating the IFN response, and overexpression of TRIM56 K110Q further enhanced the cGAS-STING-IFN axis, protecting against HSV-1 infection and HSE pathogenesis in mice. Furthermore, during the early stages of infection, HDAC6 accumulated in the cytoplasm, where it was phosphorylated by the viral protein US3. This led to an increase in deacetylase activity and interaction with TRIM56. Furthermore, the use of neurons and microglia cells allows for a more accurate modelling of the process of HSV-1 infection in brain tissue in vitro. While microglia are the primary macrophages responsible for antiviral immunity in the brain, neurons also exhibit an immune response when stimulated by external pathogenic microorganisms. The results of our study indicate that HDAC6 may regulate cGAS activity by modulating the acetylation and ubiquitination of TRIM56 in both microglia and neurons. This may lead to a reduction in the interaction between cGAS and viral DNA. Therefore, these findings highlight the role of HDAC6-mediated deacetylation as a regulatory mechanism of the TRIM56-cGAS-STING antiviral response.

Through its ubiquitinase activity and RNA binding ability, TRIM56 has been widely shown to directly target viral components to inhibit infection of several RNA viruses via its RING and C-terminal structural domains (Fu et al, 2023). Additionally, TRIM56 ubiquitinates cGAS or STING to promote innate immune responses to DNA virus infection (Seo et al, 2018; Tsuchida et al, 2010; Yang et al, 2018). While previous studies have shown that the regulatory activity of TRIM56 on the antiviral immune response is attributed to the RING domain (Fu et al, 2023; Tian et al, 2022), the function of its B-box domain remains poorly understood. In this study, we revealed that TRIM56 K110-specific acetylation within B-box1 is essential for its ubiquitination activity on cGAS and the subsequent IFN response to DNA virus infection. We showed that B-box1 overexpression can strengthen cGAS-STING-IFN axis activation, whereas knockdown of the B-box1 region significantly reduces IFN production. Additionally, HDAC6-mediated deacetylation or the substitution of K-to-A significantly abolished the B-box1-mediated increase in cGAS-STING activation and IFN production. Therefore, the K110 acetylation of TRIM56 B-box1 reinforce the intracellular DNA-mediated immune response. Furthermore, the N-terminal RD region of cGAS broadens its DNA recognition range and increases its binding efficiency of dsDNA (Wang et al, 2017). It was found that inhibition of TRIM56 K110 acetylation significantly reduces cGAS binding to DNA, implying that HDAC6-mediated deacetylation in B-box1 allosterically modulates the interaction between TRIM56 and the N-terminal RD region of cGAS (Seo et al, 2018). Although the acetyltransferases responsible for TRIM56 acetylation have not been identified, two potential TRIM56 acetylases are the histone acetyltransferases p300 and PCAF, as they have been shown to acetylate TRIM50 (Fusco et al, 2014). In addition to TRIM56, HDAC6 has been shown to regulate the activities and functions of other TRIMs, such as TRIM21, TRIM28, and TRIM50, through acetylation/deacetylation (Fusco et al, 2014; Fusco et al, 2012; Li

et al, 2021; Xie et al, 2020). For example, HDAC6 deacetylates TRIM21 at K385/K387 to stimulate its homodimerization, which in turn facilitates the proteasomal degradation of viruses bound to antibodies mediated by TRIM21 (Xie et al, 2020). HDAC6 deacetylates TRIM50 at K372, promoting its ubiquitination and subsequent transportation into the aggresome for the clearance of ubiquitinated proteins (Fusco et al, 2012). Therefore, HDAC6-mediated deacetylation is likely to be a universal regulatory mechanism controlling the ubiquitination activity of TRIMs.

HDAC6 has been extensively implicated in different stages of the viral life cycle, most notably in regulating the dynamic organization of microtubules, a critical process for viral intracellular transport (Zheng et al, 2017). Additionally, HDAC6 regulates various aspects of the immune response to DNA or RNA viruses through its deacetylase and ubiquitin-binding activity (Qu et al, 2023). For example, in the case of RNA virus infection, HDAC6 deacetylates RIG-I at K909 to enhance its ability to detect viral RNAs and to facilitate the IFN production (Choi et al, 2016; Liu et al, 2016), or deacetylates β-catenin to enhance its nuclear translocation and function as a coactivator for IRF3-mediated IFN transcription (Zhu et al, 2011). As a result, HDAC6 inhibition increases cells susceptibility to both HCV and SenV infections (Choi et al, 2016; Zhu et al, 2011). Conversely, HDAC6 promotes the autophagic degradation of K48-linked ubiquitinated cGAS to restrict the antiviral response to porcine circovirus (PCV), a DNA virus (Wang et al, 2021). In the present study, we demonstrated that, following HSV-1 infection, HDAC6 deacetylates TRIM56 at K110 to impair its ability to catalyze cGAS monoubiquitination, resulting in reduced DNA binding activity of cGAS and IFN production. Unlike PCV infection, both in vitro and in vivo results showed that HDAC6 deficiency or inhibition of its deacetylase activity did not alter cGAS protein levels. Instead, HDAC6 repressed cGAS DNA binding activity through TRIM56 deacetylation. It is worth noting that HDAC6's inhibitory effect on the TRIM56-cGAS axis did not require its ubiquitin-binding function, as demonstrated by the inhibitory effect of HDCA6-CD1/2 overexpression on cGAS-STING activation. Therefore, virus and host processes that modulate HDAC6 deacetylase activity serve to control the cGAS-STING signaling pathway.

In addition, multiple studies have shown that phosphorylation of HDAC6 affects its cellular localization and enhances its deacetylase activity (Zheng et al, 2017). Consistently, we found that the phosphorylation levels of HDAC6 are significantly increased during the early stage of HSV-1 infection, accompanied by enhanced deacetylase activity. Viral protein US3 has been shown to phosphorylate various crucial regulators, such as RIG-I, β-catenin, and IRF3, to suppress IFN-β production and promote viral infection (van Gent et al, 2022; Xie et al, 2021; You et al, 2020). Through the time-dependent HDAC6-interactome, our initial research discovered that HSV-1 US3 interacts with and phosphorylates cytoplasmic HDAC6, which leads to an increased HADC6-TRIM56 interaction and inhibits cGAS-mediated IFN production. This suggests that US3 utilizes HDAC6-mediated deacetylation as an additional strategy to counter the host antiviral immune response in microglia and neurons. Conversely, HSV-1 infection of non-neuronal cells has been observed to elicit specific and different immune evasion mechanisms (Verzosa et al, 2021). For example, in monocytes and primary macrophages, the HSV-1 UL37 deamidates cGAS, thereby impairing its ability to produce cGAMP and consequently hindering subsequent innate immunity (Zhang et al, 2018). Alternatively, in foreskin fibroblasts, HSV-1 UL41 directly degrades

cGAS mRNA (Su and Zheng, 2017). It can therefore be concluded that HSV-1 employs cell-specific strategies, via diverse viral proteins, to circumvent the cGAS-STING signaling pathway. Further investigation is recommended in order to fully assess the impact of US3 on HDAC6-mediated cGAS-STING signaling activation.

Furthermore, position 110 of TRIM56 has amino acids that differ between species, with mice having K and humans having T. The molecular dynamic stimulation assay has shown that the K-to-T substitution in human (TRIM56 T110) leads to increased binding affinity to cGAS, resulting in the enhancement of cGAS-mediated IFN production. As such, the B-box1 T110 mutant (hTRIM56) significantly enhances the activation of TBK1 and inhibits HSV-1 infection more efficiently than B-box1 K110 (mTRIM56) (Fig. EV5E,F). Consistently, the inhibitory effect of tubacin on HSV-1 infection was more observable in mouse BV2 and N2a cells (Figs. 2F and EV2D) than in human HMC3 cells (Fig. EV2E). Therefore, these findings indicate that the deficiency in TRIM56 deacetylation by HDAC6 renders the human host more resistant to HSV-1 infection. Indeed, previous studies have also demonstrated that HSV-1 exploits species-specific variation to inactivate innate defense in susceptible hosts (Huang et al, 2021; Lou et al, 2016; Zhang et al, 2018). Our study that cGAS inhibition by HDAC6-mediated deacetylation of TRIM56 constitutes a restriction mechanism in cross-specifies transmission of viruses.

In conclusion, our research shows that HDAC6 deacetylates TRIM56 at K110, leading to the suppression of cGAS-mediated innate immune signaling in a species-specific manner. The acetylation level of TRIM56 at K110 determines its interaction with HDAC6 and cGAS, resulting in contrary monoubiquitination and DNA binding activity of cGAS, ultimately affecting IFN production. Moreover, the viral protein US3 phosphorylates HDAC6 to regulate the TRIM56-mediated antiviral immune response. HDAC6 deficiency or inhibition of its deacetylase activity inhibits both HSV-1 infection and HSE pathogenesis in vitro and in vivo. Collectively, these studies reveal the crucial role played by HDAC6-mediated deacetylation in regulating the TRIM56-cGAS-mediated antiviral response, providing valuable insights for developing antiviral strategies.

## Methods

**Reagents and tools table**

| Reagent/resource | Reference or source | Identifier or catalog number |
|---|---|---|
| **Experimental models** | | |
| Mouse microglial BV2 cells | Chinese Academy of Science | 3111C0001CCC000063 |
| Human microglial HMC3 cells | American Type Culture Collection | CRL-3304 |
| mouse neuronal N2a cells | American Type Culture Collection | CCL-131 |
| Vero cells | American Type Culture Collection | CCL81 |
| HEK 293T cells | American Type Culture Collection | CRL-1573 |

| Reagent/resource | Reference or source | Identifier or catalog number |
|---|---|---|
| HMC3 TRIM56-KO12 cells | TsingKe Biological Technology | N/A |
| BALB/c mice | Experimental Animal Centre of Southern Medical University | N/A |
| HDAC6 knockout mice | Gao et al (2007) | Prof. Dr. Jun Zhou, Nankai University, China |
| HSV-1 strain F | Li et al (2022) | VR-733 |
| EGFP-tagged HSV-1 strains | Li et al (2022) | Prof. Dr. Kurihara Hiroshi, Jinan University, China |
| **Recombinant DNA** | | |
| pCMV-HA-C | This study | N/A |
| pCMV-HA-C-HDAC6 | This study | N/A |
| pCMV-HA-C-HDAC6-ΔNLS | This study | N/A |
| pCMV-HA-C-HDAC6-CD1/2 | This study | N/A |
| pCMV-HA-C-HDAC6-HD1/2 m | This study | N/A |
| pCDNA3.1-Flag | This study | N/A |
| pCDNA3.1-Flag TRIM56 | This study | N/A |
| pCDNA3.1-Flag TRIM56 K110Q | This study | N/A |
| pCDNA3.1-Flag TRIM56 K110R | This study | N/A |
| pCDNA3.1-Flag B-box1 | This study | N/A |
| pCDNA3.1-Flag B-box1 K110A | This study | N/A |
| pCDNA3.1-Flag B-box1 K110T | This study | N/A |
| **Antibodies** | | |
| Anti-HDAC6 (WB, IP) | Cell Signaling Technology | 7612 |
| Anti-α-tubulin | Cell Signaling Technology | 3873 |
| Anti-acetyl-α-tubulin | Cell Signaling Technology | 5335 |
| Anti-cGAS | Cell Signaling Technology | 31659 |
| Anti-phospho-TBK1 (P-TBK1) | Cell Signaling Technology | 5483 |
| Anti-TBK1 | Cell Signaling Technology | 3504 |
| Anti-phospho-IRF3 (p-IRF3) | Cell Signaling Technology | 4947 |
| Anti-IRF3 | Cell Signaling Technology | 4302 |
| Anti-phospho-STING (p-STING) | Cell Signaling Technology | 72971 |
| Anti-STING | Cell Signaling Technology | 13647 |

| Reagent/resource | Reference or source | Identifier or catalog number |
|---|---|---|
| Anti-GAPDH | Cell Signaling Technology | 2118 |
| Anti-β-actin | Cell Signaling Technology | 4970 |
| Anti-Histone H3 | Cell Signaling Technology | 4499 |
| Anti-HA-tag | Cell Signaling Technology | 3956 |
| Anti-DYKDDDK-Tag | Cell Signaling Technology | 70569 |
| Anti-TRIM56 | SAB signalway Antibody | 30308 |
| Anti-phosphoserine | Abcam | ab7851 |
| Anti-HSV-1 | Abcam | ab9533 |
| Anti-ICP0 | Abcam | ab6513 |
| Anti-gD | Abcam | ab6507 |
| Anti-acetyl-lysine | Jingjie PTM BioLab | PTM-105 |
| Anti-HDAC6 (IF) | Abmart | PS02180S |
| Anti-Ubiquitin | LifeSensors | LSS-VU-0101-0050 |
| Goat anti-rabbit H&L | ZEN-BIOSCIENCE | 511203 |
| Goat anti-mouse H&L | ZEN-BIOSCIENCE | 511103 |
| **Oligonucleotides and other sequence-based reagents** | | |
| $U_L27$ | This study | F: 5′-GCCTTCTTCGCCTTTCGC-3′ R:5′-CGCTCGTGCCCTTCTTCTT-3′ |
| $U_L52$ | This study | F: 5′-GACCGAGGGTGCGTTATT-3′ R:5′-GAAGGATCGCCATTTAGCC-3′ |
| $U_L54$ | This study | F:5′-GAAGGATCGCCATTTAGCC-3′ R:5′-TGGCCGTCAACTCGCAGA-3 |
| $U_L47$ | This study | F:5′-ACGATGATGATGAGGTTCCC-3′ R:5′-ACGATGATGATGAGGTTCCC-3 |
| Ifnb1 (mouse) | This study | F:5′-ATGAGTGGTGGTTGCAGGC-3′ R:5′-TGACCTTTCAAATGCAGTAGATTCA-3 |
| Ifnα4(mouse) | This study | F:5′-TGATGAGCTACTACTGGTCAGC-3′ R:5′-TGATGAGCTACTACTGGTCAGC-3 |
| Cxcl10(mouse) | This study | F:5′-CCAAGTGCTGCCGTCATTTTC-3′ R:5′-GGCTCGCAGGGATGATTTCAA-3 |
| Isg15(mouse) | This study | F:5′-GATTGCCCAGAAGATTGGTG-3′ R:5′-TCTGCGTCAGAAAGACCTCA-3 |
| Mx2(mouse) | This study | F:5′-GAGGCTCTTCAGAATGAGCAAA-3′ R:5′-CTCTGCGGTCAGTCTCTCT-3 |
| TRIM56 (mouse)(ASO) | This study | F:5′-GTATGATGAAGAGGCTCGAGAG-3′ R:5′-CAACTGAGAACAAGGTTGACAG-3 |
| HSV-1 US3 | This study | F:5′-GGAGATCCTTGCCCAGATGT-3′ R:5′-CTCTGCGTATTCCTCCGGAT-3′ |
| HDAC6 (mouse) | This study | F:5′-ACGCTGACTACATTGCTGCT-3′ R:5′-ATCTCTCCCTTGGGGTCTCC-3′ |
| β-actin (mouse) | This study | F:5′-GGCTTATTCCCCTCCATCG-3′ R:5′-CCAGTTGGTAACAATGCCATGT-3′ |
| EGFP | This study | F:5′-AGGAGCGCACCATCTTCTTC-3′ R:5′-CTCGATGTTGTGGCGGATCT-3′ |
| **Chemicals, enzymes and other reagents** | | |
| Tubacin | TOPSCIENCE | T6327 |
| ISD Naked | InvivoGen | tlrl-isdn |
| STING agonist (2′3′-cGAMP) | InvivoGen | tlrl-nacga23 |

| Reagent/ resource | Reference or source | Identifier or catalog number |
|---|---|---|
| CT-DNA | Solarbio | D8020 |
| **Software** | | |
| GraphPad Prism 8 | https://www.graphpad.com | |
| ZEN software | https://www.zeiss.com.cn/corporate/home.html | |
| ImageJ | https://imagej.net/ij/index.html | |
| NIS-Elements | https://www.microscope.healthcare.nikon.com | |
| Quantity One | https://www.bio-rad.com/en-us/product/quantity-one-1-d-analysis-software?ID=1de9eb3a-1eb5-4edb-82d2-68b91bf360fb | |
| **Other** | | |
| DAPI | Beyotime | C1006 |
| Flag magnetic beads | Beyotime | P2181 |
| HA magnetic beads | Beyotime | P2185 |
| DMEM | GIBCO | 8118305 |
| FBS | ExCell Bio | FND500 |

## Cell line culture and viruses

Mouse microglial BV2 cells, mouse neuronal N2a cells (Cat# CCL-131), Vero cells (Cat# CCL81) and HEK 293 T (Cat# CRL-11,268) were cultured in DMEM (GIBCO, 8118305, Carlsbad, California, USA) supplemented with 10% FBS (ExCell Bio, FND500, Shanghai, China), and then incubated at 37 °C with 5% CO2. HSV-1 strain F and Enhanced Green Fluorescent Protein (EGFP)-tagged HSV-1 strains were prepared and viral stocks were titrated by plaque assays in Vero cells (Li et al, 2022).

## Animal studies

BALB/c mice (aged 5-6 weeks) were obtained from the Experimental Animal Centre of Southern Medical University (Guangzhou, China). Ethical approval for mouse studies was granted by the Jinan University Institutional Animal Care and Use Committee (No. 20210829-04). The mice were given one week to adapt prior to the experiment. The lentiviruses employed in this investigation were created, synthesized, and purified by TsingKe Biological Technology (Beijing, China). All the titers of lentivirus used in this study were higher than $10^8$ transduction units (TU) per mL to ensure superior transduction efficiency (Wang et al, 2022). The HSE mouse model was established and the pathological scores were recorded as previously described (Li et al, 2022; Wang et al, 2022). Briefly, mice received intranasal inoculation of HSV-1 at $2 \times 10^6$ PFU/mouse. Daily monitoring of body weight and assessment of HSE symptoms associated with neurological disease were conducted.

HDAC6 knockout (KO) mice were kindly provided by Jun Zhou from Nankai University, China. As HDAC6 gene is located on the X chromosome, male mice with the mutant allele are HDAC6-null (Gao et al, 2007). Subsequently, we intercrossed mice to produce both WT and HDAC6 KO mice. Genomic DNA was extracted from mouse tails for genotyping (#D7283S, Beyotime). Primers for HDAC6 genotyping of mice: Zeo-1:5'-CCA TGA CCG AGA TCG GCG AGC A-3', Zeo-3: 5'-CGT GAA TTC CGA TCA TAT TCAAT-3'. All experiments were conducted using HDAC6 KO and wild-type (WT) male mice aged 6–8 weeks.

## HDAC6 enzyme activity determination

N2a cells were infected with or without HSV-1 (MOI = 5) for 0, 2, 4, 6 h or treated with tubacin (1 μM) alone for 2 h. HDAC6 enzyme activity was determined following the guidelines of the HDAC6 Activity Assay Kit (Fluorometric) provided by Abcam (ab284549).

## Western blot analysis

Cells were lysed using SDS lysis buffer (P0013G, Beyotime, Shanghai, China) containing 1% PMSF (ST506, Beyotime) and 1% phosphatase inhibitors (ST650, Beyotime). The protein concentrations were then determined using the BCA protein assay kit (P0009, Beyotime). Total proteins were separated using sodium dodecyl sulfate polyacrylamide gel electrophoresis (SDS-PAGE) and then electro-transferred to PVDF membranes (ISEQ00010, Millipore, USA). The membranes were blocked with 5% skim milk for 1 h at room temperature, and then incubated overnight at 4 °C in the presence of specific antibodies. The samples were subsequently incubated with HRP-linked secondary antibodies (Luan et al, 2023). β-actin was used as a loading control. The protein bands were ultimately visualized using enhanced chemiluminescence. The protein bands were subsequently subjected to quantitative analysis using Quantity One.

## RNA isolation and quantitative real-time PCR (RT-qPCR)

Total RNA was extracted using TRIzol Reagent (TIANGEN, #DP405). For cDNA synthesis, one microgram of RNA per sample was utilized with the PrimeScript RT Reagent and the gDNA Eraser Kit (Takara, #RR047A). RT-qPCR assays were carried out following our published methods (Li et al, 2022; Wang et al, 2022). Gene expression levels were standardized to the internal housekeeping gene β-actin. All qPCR procedures, including primer design, PCR condition validation and quantification, were conducted in line with MIQE guidelines. The gene-specific primers are provided in the "Reagents and Tools Table".

## SiRNA and ASO transfection

All small interfering RNAs (siRNAs) targeting HDAC6 (1#: sense: 5'-CCU UGA AGC GUG GUG GAA ATT-3', antisense: 5'-UUU CCA CCA CGC UUC AAG GTT-3'; 2#: sense: 5'-GCU UCU AAC UGG UCC ACU ATT-3', antisense: 5'-UAG UGG ACC AGU UAG AAG CTT-3'; 3#: sense:5'-CCU GGA CUC UGA GCU CCU UTT-3', antisense: 5'-AAG GAG CUC AGA GUC CAG GTT-3') or TRIM56 (1#: sense: 5'-CCA UGC CUA CAU ACC UAU UTT-3', antisense: 5'-AAU AGG UAU GUA GGC AUG GTT-3'; 2#: sense: 5'-GCA GAA CAG GGU CUU GAA ATT-3', antisense: 5'-UAG UUU CAA GAC CCU GUU CUG CTT-3'; 3#: sense:5'-CCC UUC GAG AGG UAA ACA ATT-3', antisense: 5'-UUG UUU ACC UCU CGA AGG GTT-3') were designed, synthesized, and purified by GenePharma (Beijing, China). Antisense oligonucleotides (ASOs) targeting TRIM56 were designed, synthesized, and purified by TsingKe Biological Technology (Beijing, China). The sequences of ASOs targeting TRIM56 B-box1, which were used in our experiments, are listed below. ASO-TRIM56 B-box1 1#: 5'-AAG TCG TCG GCA CAG TCC AG-3', ASO-TRIM56 B-box1 2#: 5'-AAT CCA AGT CAA CGA CAC GG-3', ASO-TRIM56 B-box1 3#:

5'-ACT TGT CAT CGT CGT CCT TG-3'. For siRNA transfection, the INTERFERin reagent (#PT-409-10, Polypus Transfection, USA) was used following the manufacturer's instructions. Briefly, BV2 or N2a cells were cultured in six-well plates overnight at 37 °C. NC siRNA and HDAC6 or TRIM56 siRNA were added to the corresponding wells at a final concentration of 100 nM. After 6 h of transfection, the cells were washed with PBS and cultured in fresh DMEM medium containing 10% FBS for 72 h. The cells were subsequently used for subsequent experiments.

## Construction and transfection of the plasmid

Full-length wild-type HDAC6 (NM_010413.3) and mutants HDAC6-△NLS (deficiency of nuclear localization sequence), HDAC6-CD1/2 (a truncated deacetylase active domain), and HDAC6-HD1/2 m (both catalytic domains contain inhibitory mutations, D250N/D252N/H254V/H255V in catalytic core HD1m; D648N/H650N in HD2m) were inserted into the pCMV-HA-C vector. Full-length wild-type TRIM56 (NM_201373.4) and its mutant with space (pCDNA3.1-Flag) were synthesized and purified by TsingKe Biological Technology (Beijing, China). pGPU6/GFP/Neo-HDAC6-Mus: 5'GCTTCTAAC TGGTCCAC TA-3' was purchased from GenePharma (Shanghai, China). The cells cultured in 6-well plates were added to the corresponding plasmids (4 μg) using jetPEI reagents (Polyplus Transfection, IIIkirch, France, PT-114-15). After 6 h of transfection, the cells were washed using PBS and cultured in fresh DMEM containing 10% FBS for 72 h before being utilized for subsequent experiments.

## Viral plaque assay

N2a, BV2 and HMC3 cells were treatment with HSV-1 and tubacin or transfection of the siRNA or plasmid. The samples were collected and stored at −80 °C for future use after undergoing the process of being repeatedly frozen and thawed at −80 °C for three times. A viral plaque reduction assay was conducted on vero cells as previously described (Wang et al, 2023). Briefly, the cells grown in 24-well plates were exposed to virus for absorption and the culture medium was replaced by maintenance medium supplemented with 1% methylcellulose (T10218-500, Sijia Biotech, Guangzhou, China). The cells were then incubated for 72 h before fixation using 4% polyoxymethylene and staining with 1% crystal violet (C0121, Beyotime). Afterwards, the total number of plaques was counted to enable statistical analysis.

## Viral DNA purification and quantification

Viral DNA isolation and quantification were performed based on our prior study (Wang et al, 2023). Briefly, cells with the corresponding treatment were repeatedly frozen at −80 °C and thawed three times. Afterward, viral DNA was isolated using the TIANamp Virus DNA/RNA kit (Transgene, Beijing, China, #ER201-01). The purified viral DNA was quantified using a RT-qPCR assay. The internal housekeeping gene β-actin or the exogenous gene EGFP was used as a control (In the viral DNA isolation experiments of animal and IP-cGAS experiments, exogenous EGFP was supplemented and as a control).

## Enzyme-linked immunosorbent assay (ELISA)

The concentrations of IFN-β in culture supernatants and sera were measured by ELISA kits (4 A Biotech Co., Ltd, CME0116, Beijing, China).

## Immunoprecipitations

A portion of the supernatant (50 μl) was retained as total cell lysate, and the remainder was incubated overnight with an indicated antibody (or an equal amount of IgG as a control) and protein A/G magnetic beads (B23202, Bimake). For IP with HA magnetic beads (P2185S, Beyotime), cell lysates were incubated with the beads overnight. The immunoprecipitates were washed three times with IP lysis buffer. Proteins were eluted from the beads by boiling in 60 μL SDS loading buffer and then analyzed by western blotting.

## Immunofluorescence assay

N2a cells were seeded in glass-bottomed dishes and infected with HSV-1 (MOI = 5) for the indicated time. The supernatants were removed and the cells were washed with PBS, fixed with 4% PFA, permeabilized with 0.2% Triton X-100 and blocked in 3% BSA for 1 h. The cells were then incubated with HDAC6 antibodies (1:300) overnight at 4 °C and then with fluorescent secondary antibodies for 1 h at room temperature. Finally, the cells were incubated with DAPI for 15 min at room temperature. Confocal pictures were captured using a Zeiss AxioCam MR R3 cooled CCD camera controlled by ZEN software (Carl Zeiss MicroImaging GmbH, Göttingen, Germany).

## Nuclear and cytoplasmic extract preparation

N2a cells were seeded in 10 cm dishes and infected with HSV-1 (MOI = 5) for the indicated time (0 h, 2 h, 4 h, 6 h). The supernatants were removed and the cells were washed with PBS, and then the cells were re-suspended with 5 ml PBS and transferred to a new centrifuge tube. After centrifugation at 4 °C, 1000 rpm for 5 min, PBS was absorbed and finally precipitated cells were obtained. The preparation of nuclear and cytosolic proteins was performed using a commercial Nuclear and Cytoplasmic Protei-nExtraction Kit (P0028, Beyotime, China).

## Histopathological and immunohistochemical assays

All organs of the mice, including the brain, liver, kidney, and spleen, were harvested, fixed in 4% PFA overnight and embedded in paraffin. The paraffin blocks were then sectioned and stained with hematoxylin and eosin (H&E). For immunohistochemistry, the brains of HDAC6 KO mice were stained with anti-HDAC6. All images were acquired using NIS-Elements Viewer.

## Proteomic assay of HDAC6 interactome and acetylome

For acetylome analysis, proteins were digested using filter-assisted sample preparation (FASP). The resulting peptides were subjected to acetylated peptide enrichment according to the manufacturer's

instructions (PTM-104, Biolab, China). The peptide mixtures were further used for data-dependent acquisition (DDA) mass spectrometry analysis, data processing, and differential expression analysis of the acetylomic (further experimental details are provided in the supplementary information). For HDAC6 interactome identification, N2a cells were transfected with HA-HDAC6 or HA-vector (15 μg) for 48 h and cells were harvested with NP-40 lysis buffer (P0013F, Beyotime, China) containing protease inhibitor cocktail tablets (Roche Diagnostics GmbH, Mannheim, Germany). The lysate was subjected to HA-antibody immunoprecipitation, and the resulting pellet was eluted using 0.3% formic acid. The collected eluate was then digested following the FASP protocol. Subsequently, purified peptides were split for data-independent acquisition (DIA) mass spectrum profiling, data processing, WGCNA and functional enrichment analysis (further experimental details are provided in the supplementary information).

## Molecular docking and dynamics simulation

The crystal structures of catalytic domain 1 (CD1) and catalytic domain 2 (CD2) of HDAC6 (PDB ID: 5g0i) and the B-box1 domain of murine TRIM28 were employed as templates for the murine HDAC6 CD1-CD2 and TRIM56 B-box1 domains, respectively. Subsequently, the protein structures were constructed utilizing the homology modeling algorithm in Modeller 9.23 software. We used the I-TASSER online server to produce the 3D structures of both full-length cGAS and TRIM56, which lack available crystalline structures and homologous templates. They maintained the default settings for all parameters and selected the top-scoring structure as the molecular docking receptor. PyMOL was utilized to incorporate hydrogen atoms, following which the resulting structure was saved as a PDB file for use as the protein docking ligand. The HDAC6 model served as the receptor while the TRIM56 B-box1 domain or TRIM56 Bbox1 K110A/T domain was used as a ligand during protein-protein docking via the Z-dock server (https://zdock.umassmed.edu/). Default parameters were utilized, and the molecular docking pose with the highest score was selected (further details are provided in the supplementary information).

## Acetylome

### Protein digestion and acetylated peptides enrichment

Proteins were digested by the Filter-aided sample preparation (FASP) (Wiśniewski et al, 2009). In brief, four milligrams of proteins were transferred to an ultrafiltration tube (Ultracel-10K, Millipore, USA). The protein samples were reduced with a 20 mM dithiothreitol (DTT) for 1.5 h at 37 °C, and then alkylated with 100 mM iodoacetamide (IAA) for 20 min at RT in the dark. Subsequently, the samples underwent digestion with Trypsin (at a ratio of 1:100, W/W, V5111, Promega, USA), and the resulting peptides were subject to enrichment of acetylated peptides according to the manufacturer's instructions (PTM-104, Biolab, China). The peptide mixtures were employed for Data-Dependent Acquisition (DDA) profiling of the acetylome.

### DDA MS analyses and data processing

The enriched acetylated peptide samples were separated by an EasyNano LC1000 system (San Jose, Thermo Fisher) using a C18 column (3 μm, 75 μm × 15 cm) at a 500 nl/min flow rate. A 75-min

gradient was set as follows: 1% B (0.1% FA in ACN) /99% A (0.1% FA in H2O) to 3% B in 2 min, 3% B to 8% B in 8 min, 8% B to 20% B in 45 min, 20% B to 30% B in 12 min, 30% B to 90% B in 1 min and kept for 7 min. MS data were acquired with a data-dependent acquisition mode using Orbitrap Fusion Lumos (Bremen, Thermo Fisher). For the data acquisition, a fullspeed scan mode of 3 s with an MS1 scan range of $m/z$ 350–1550 was used, and other parameters were set as below: MS1 and MS2 resolution was set to 120 K and 30 K; Automatic gain control (AGC) was used to prevent overfilling of the ion trap, for MS1 and MS2 was 1e6 and 5e4; isolation window was 1.6 $m/z$, higher energy C-trap dissociation (HCD) with normalized collision energy (NCE) was 32, dynamic exclusion time was 20 s. Raw MS data were searched against a Mus musculus protein database (Uniprot: Swiss-Prot, 17001 entries, acquired on 2019.01.09) using Maxquant (version 1.6.3.3). The following parameters were used for data processing: trypsin, with a maximum number of 3 missed cleavages; precursor and fragment ion mass tolerance was set to 10 ppm; variable modification was set to Oxidation on methionine (M, 15.9949) and acetylation on lysine (K, 42.0106) and protein N-terminus; fixed modification was set to Carbamidomethylation on cysteine (C, 57.0215); An algorithm of Percolator was used to keep peptide FDR less than 1% and the $q$ value used for protein identification was less than 0.01. Modification probability was kept at more than 0.75.

### Differential expression analysis of acetylomic

To acquire the relative quantitation of acetylated sites in each sample. The intensity of detected acetylated sites in the sample was adjusted using the total ion current intensity to derive site-relative quantitative values. These relative quantitative values were subsequently employed for the analysis of differential expression of acetylated sites using Perseus software (version 1.6.6.0) (Tyanova and Cox, 2018). The statistical significance of differential expression of acetylated sites and proteins were perfomed by Student's T-test and the p-value corrected FDR ($q$ value) was acquired by Benjamini-Hochberg. For acetylomic |log2(fold change)|≥0.263 $P$ value of acetylated sites were setted <0.05 to detect differential expressed site (DES).

## HDAC6-interactome

### DIA MS analyses and data processing

The freeze-dried tryptic peptide was reconstituted by deionized water containing 0.1% (v:v) formic acid. To allow retention time calibration, the iRT-standard provided by the iRT-Kit (Biognosys, Schlieren, Switzerland) was added into each sample at 1/10 by volume. Samples were then analyzed in a DIA mode by the LC–MS/MS, equipped with an EASY-nLC 1200 (Thermo) HPLC system and Orbitrap Fusion Lumos (Thermo) mass spectrometer. For LC separation, tryptic peptides were sequentially injected into an Acclaim PepMap 100 C18 column (100 μm × 2 cm, 5 μm, Thermo, P/N: 164564) and an Acclaim PepMap 100 C18 column (50 μm × 15 cm, 2 μm, Thermo, P/N: 164943). Mobile phases included Buffer A [0.1% (v: v) formic acid] and Buffer B (80% acetonitrile with 0.1% formic acid). With a flow rate of 600 nL/min, the gradients included: 15 min equilibration with 100% of Buffer A; Buffer B, 7–12%, 0–12 min; Buffer B, 12–30%, 12–79 min; Buffer B, 30%–40%, 79–104 min; Buffer B, 40%–95%, 104–105 min; wash with 95% Buffer B, 15 min. The parameters are listed as follows: ion

source type, nanospray ionization; spray voltage, 2.1 kV (positive); ion transfer tube temp, 320 °C; default charge state, 2; full-MS scan in the scan cycle included 37 of MS/MS scans. MS parameters: detector type, orbitrap; MS resolution, 60,000 at 400 $m/z$; mass range, normal; scan range, 350–1200 $m/z$; S-lens RF level, 30%; MS AGC target, 4.0e5; maximum injection time, 30 ms. MS/MS parameters: MS/MS isolation mode, quadrupole; activation type, HCD; HCD collision energy, 30%; detector type, orbitrap; MS/MS resolution, 30,000 at 400 $m/z$; mass range, normal; scan range, 200–2000 $m/z$; MS/MS AGC target, 5.0e5; maximum injection time, 50 ms; enabling of inject for all parallelizable time, no; enabling of MS/MS centroiding, no.

Spectronaut 14.10 (Biognosys, Schlieren, Switzerland) was used for data analysis of the raw files. All runs of the samples were searched against the Mus musculus protein database (Uniprot: Swiss-Prot, 17001 entries, acquired on 2019.01.09) and all Human herpesvirus 1 proteins sequence database, using the directDIA function in Spectronaut 14.10. Analysis against the spectral library was done in the Analysis Perspective of Spectronaut by uploading all raw files, assigning the spectral library to each file and applying the Biognosys (BGS) Factory Settings. Details are listed as follows. (1) Calibration: calibration mode, automatic; iRT calibration strategy, nonliner iRT calibration. (2) Identification: $P$ value estimator, kernel density estimator; precursor q-value cutoff, 0.01. (3) peptides length: 6. In the report section of Spectronaut, protein report files containing the protein identification and quantitation information were exported for SAINTexpress (version: v3.6.3) software to calculated HDAC6 interaction score. Proteins that fulfilled filtering criteria in SAINTexpress BFDR ≤ 0.05 and SaintScore ≥0.7 were considered to be high-confidence protein–protein interactions and visualized with Cytoscape (v.3.3.0) (Shannon et al, 2003).

### Weighted gene coexpression network analysis (WGCNA)

WGCNA is a powerful method in bioinformatics and systems biology for finding coexpression patterns in protein or gene data. In this paper, we applied WGCNA method to acetylomic. The WGCNA R package was used to construct the coexpression network. First, the samples underwent clustering to identify potential outliers. Subsequently, the coexpression network was constructed using the automatic network construction function. The R function "pickSoftThreshold" was employed to determine the appropriate soft thresholding power, which is applied to elevate the coexpression similarity for adjacency calculation. Next, hierarchical clustering and the dynamic tree-cutting function were applied to identify modules. Gene significance (GS) and module membership (MM) were computed to distinguish the vital modules associated with different time points of virus infection. Relevant gene information from the respective modules was extracted for subsequent analysis. The modules of interest were subjected to hub node identification using the Maximal Clique Centrality (MCC) algorithm within the Cytoscape software, with the assistance of the cytoHubba plugin.

### Functional enrichment analysis

The proteins interacting with HDAC6 and the proteins related to the Turquoise module identified by WGCNA were subjected to functional enrichment analysis using the R package ClusterProfile (v3.14.3) [clusterProfiler: an R package for comparing biological themes among gene clusters]. Mouse reactome pathway database was acquired by R pakage msigdbr (version: 7.4.1).

### Molecular dynamics simulation

Molecular dynamics simulations were executed on protein-protein complexes obtained via molecular docking, using the Amber18 software. Parameters for both the protein receptor and ligand were established using the ff14SB force field. The protein–protein complex was loaded into the tleap module, automatically adding hydrogen atoms and counter ions for charge neutralization. TIP3P explicit water model was applied and periodic boundary conditions were set. The molecular dynamics simulation process consisted of energy minimization, heating, equilibration, and production dynamics. Heavy atoms of the protein were initially constrained, and a 10,000-step energy minimization was performed, comprising 5,000 steps of steepest descent and 5000 steps of conjugate gradient minimization. Subsequently, the constraints were released, and a 10,000-step energy optimization for the entire system was performed. Following an initial energy minimization, the system was subsequently heated to 300 K over a period of 50 picoseconds, before undergoing a 50 picosecond equilibration under NPT ensemble conditions. A 100 nanosecond molecular dynamics simulation was performed under NPT ensemble conditions, with a time step of 2 femtoseconds. Trajectory data was saved every 20 picoseconds, before subsequent analysis using the CPPTRAJ module. Binding free energy between the ligand and receptor was then calculated using the MMPBSA.py module.

### Statistical analysis

All statistical analyses were conducted using GraphPad Prism 8 software, with statistical significance set to *$P < 0.05$, **$P < 0.01$, ***$P < 0.001$. The appropriate method of statistical analysis, was utilized as specified in the figure legends.

# Data availability

The mass spectrometry data were deposited to the Proteome Xchange Consortium via the PRIDE partner repository with the dataset identifier PXD046808 and will be available at the following link: https://www.ebi.ac.uk/pride/archive/projects/PXD046808.

The source data of this paper are collected in the following database record: biostudies:S-SCDT-10_1038-S44319-024-00358-5.

# Peer review information

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

## Acknowledgements

This work was supported by the Basic and Applied Basic Research Foundation of Guangdong Province (2021B1515120088, 2023A1515111037), the National Natural Science Foundation of China (82372244, 82373917, 82072274, 82273040, 82404704), the Modern Agricultural Industry Technology System Innovation Team Project of Guangdong Province (2020KJ142, 2021KJ142), the Guangzhou Key Research and Development Program (202206010008), the Key Basic Research Project of Shenzhen (JCYJ20220818102605011), and the Science and Technology Foundation of Shenzhen (JCYJ20200109144410181) and the Five Three Program of Shenzhen People's Hospital (SYWGSJCYJ202404).

## Author contributions

**Qiongzhen Zeng**: Conceptualization; Formal analysis; Validation; Visualization; Writing—original draft. **Zixin Chen**: Formal analysis; Visualization; Writing—original draft. **Shan Li**: Validation. **Ziwei Huang**: Validation. **Zhe Ren**: Project administration. **Cuifang Ye**: Project administration. **Xiao Wang**: Resources; Project administration. **Jun Zhou**: Resources. **Kaisheng Liu**: Resources; Supervision; Writing—review and editing. **Kai Zheng**: Supervision; Writing—original draft; Writing—review and editing. **Yifei Wang**: Resources; Supervision; Writing—original draft; Writing—review and editing.

Source data underlying figure panels in this paper may have individual authorship assigned. Where available, figure panel/source data authorship is listed in the following database record: biostudies:S-SCDT-10_1038-S44319-024-00358-5.

## Disclosure and competing interests statement

The authors declare no competing interests.

# Expanded View Figures

**Figure EV1.  HDAC6 negatively regulates IFN production.**

(**A**, **B**) N2a cells were transfected with negative control (NC) or HDAC6 siRNAs for 72 h. Knockdown efficacy of HDAC6 siRNAs in N2a cells (**A**). RT-qPCR analysis of the mRNA expression of *Ifnb1*, C*xcl10* and *Isg15* in N2a cells transfected with DHAC6 siRNAs for 72 h followed by infection with HSV-1 (MOI = 3) for 12 h (**B**). Data were analyzed using the unpaired *t* test, which are shown as mean ± SD (*n* = 3 biological replicates). (**C**) Western blot analysis of the indicated proteins in BV2 cells infected with HSV-1 (MOI = 3) for different time points. (**D–F**) N2a cells were transfected with vector, HDAC6 WT or deacetylase activity mutation (HD1/2 m) for 72 h followed by infection without or with HSV-1 (MOI = 3) for 12 h. (**D**, **E**) Western blot analysis of the indicated proteins in cells. (**F**) The mRNA expression of *Isg15*, *Isg20* and *Mx2* was detected by RT-qPCR assay. Data were analyzed using the unpaired t-test, which are shown as mean ± SD (*n* = 3 biological replicates). (**G**, **H**) Effect of tubacin on IFN production. N2a cells (**G**) and BV2 cells (**H**) were infected with HSV-1 (MOI = 3) in the presence or absence of tubacin (1 μM) for 12 h, and RT-qPCR analysis was performed to detect the expression of *Ifnb1*, *ifna4*, *cxcl10*, *isg15* genes. Data were analyzed using the unpaired t-test, which are shown as mean ± SD (*n* = 3 biological replicates). (**I**) Western blot analysis of tubacin's effect on TBK1 activation. Source data are available online for this figure.

▶

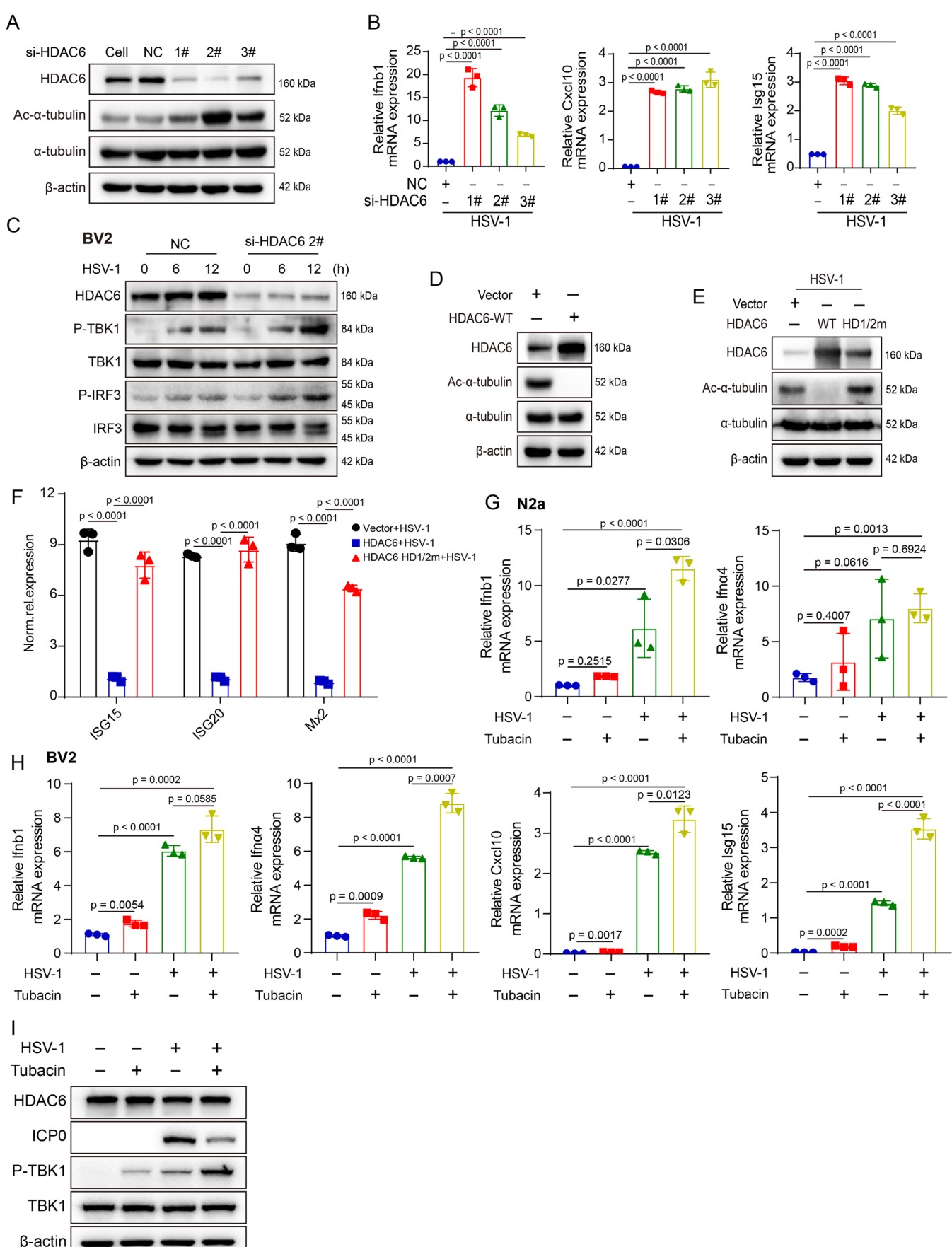

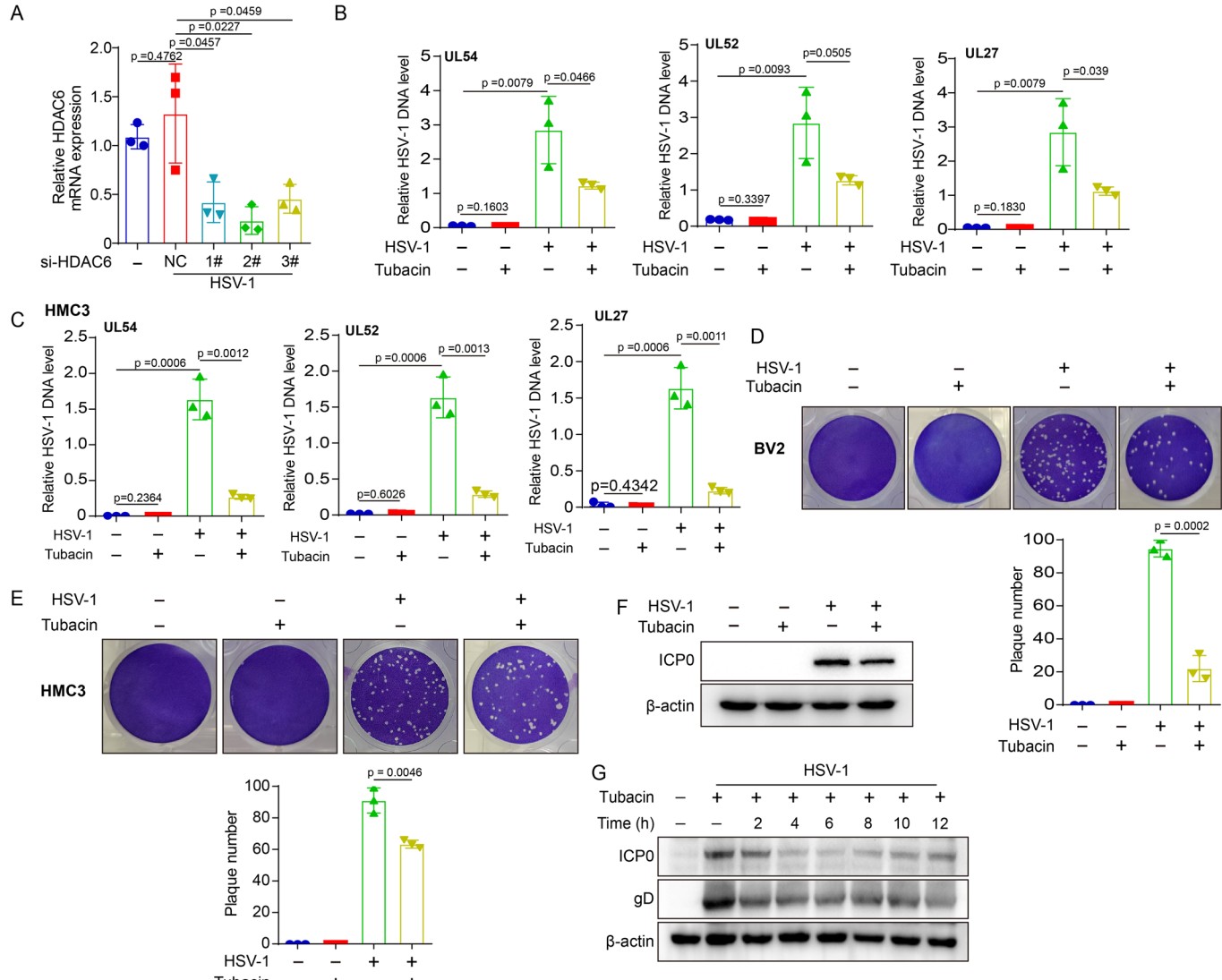

**Figure EV2. HDAC6 promotes HSV-1 infection.**

(A, B) RT-qPCR analysis of the mRNA expression of *HDAC6* in N2a cells transfected with HDAC6 siRNA and infected with HSV-1 (MOI = 3) for 12 h. Data were analyzed using the unpaired *t* test, which are shown as mean ± SD (*n* = 3 biological replicates). (B, C) The DNA copy number of viral genes in BV2 (B) and HMC3 cells (C) infected with HSV-1 (MOI = 3) in the presence or absence of tubacin (1 μM) for 12 h. Data were analyzed using the unpaired *t* test, which are shown as mean ± SD (*n* = 3 biological replicates). (D, E) Viral plaque assays of BV2 (D) and HMC3 cells (E) treated with or without HSV-1 (MOI = 3) in the presence or absence of tubacin (1 μM) for 12 h. The number of plaques was calculated. Data were analyzed using the unpaired *t* test, which are shown as mean ± SD (*n* = 3 biological replicates). (F, G) Western blot assay was performed to assess the levels of ICP0 and gD in N2a cells infected with HSV-1 (MOI = 3) for 12 h followed by treatment with tubacin (1 μM) in the indicated times of HSV-1 infection. Source data are available online for this figure.

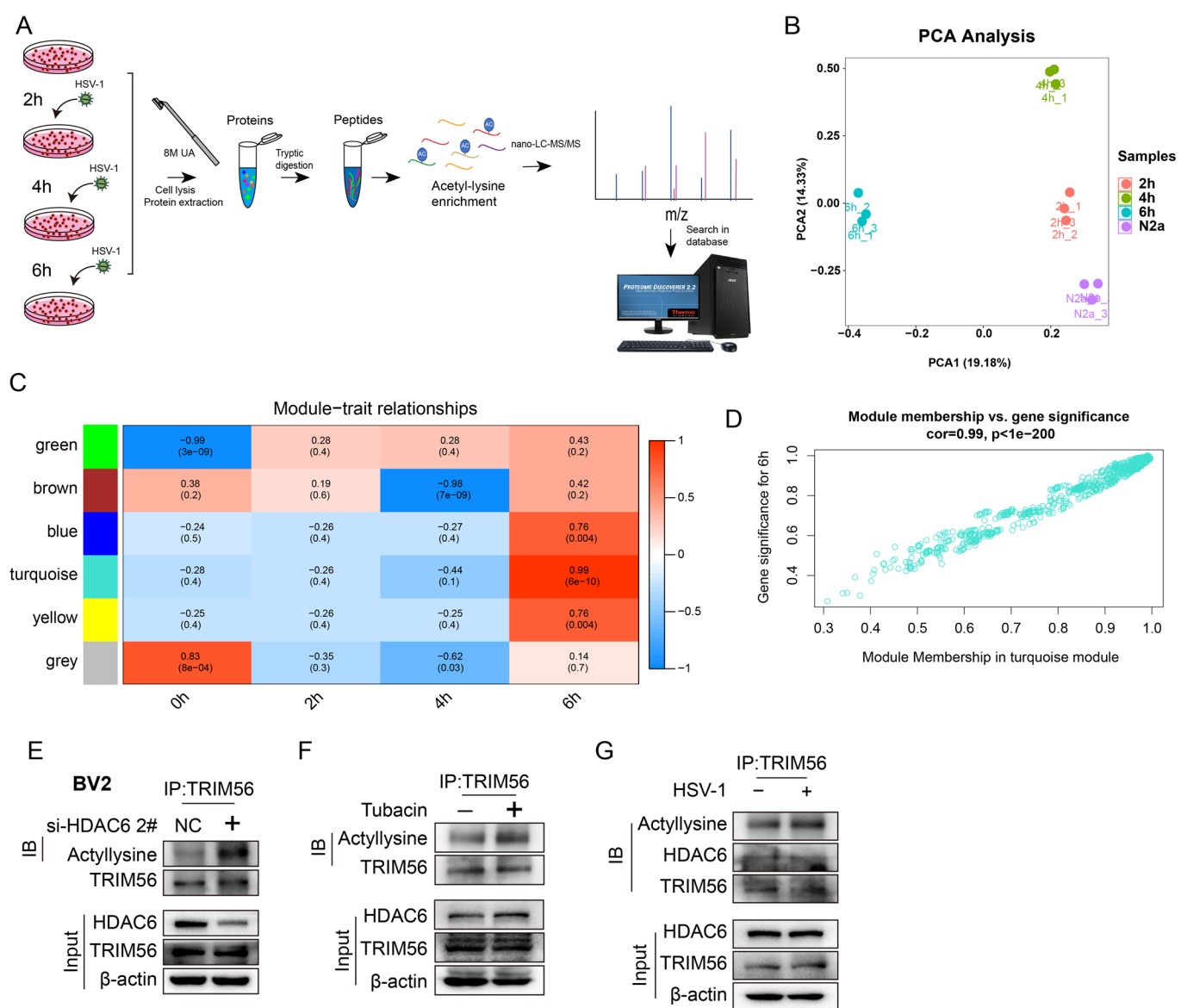

**Figure EV3.  HDAC6 regulates TRIM56 deacetylation in early stages of HSV-1 infection.**

(A) Workflow diagram illustrating the process of profiling various time points in the acetylome of N2a cells infected with HSV-1 (MOI = 10). (B) PCA analysis of each sample. (C) Each row corresponds to a module feature point, and each column corresponds to the HSV-1 (MOI = 10) infected time point. The right panel represents the range of correlations: red indicates positive correlations and blue indicates negative correlations; numbers indicate correlations and significance. (D) Correlations between sites within the turquoise module and highly correlated sites following a 6 h infection. (E–G) Immunoprecipitation analysis of the acetylation of TRIM56 in BV2 cells transfected with or without HDAC6 siRNA for 72 h (E), and treated with or without tubacin (1 μM) (F), and infected with HSV-1 (MOI = 3) for 12 h (G). Source data are available online for this figure.

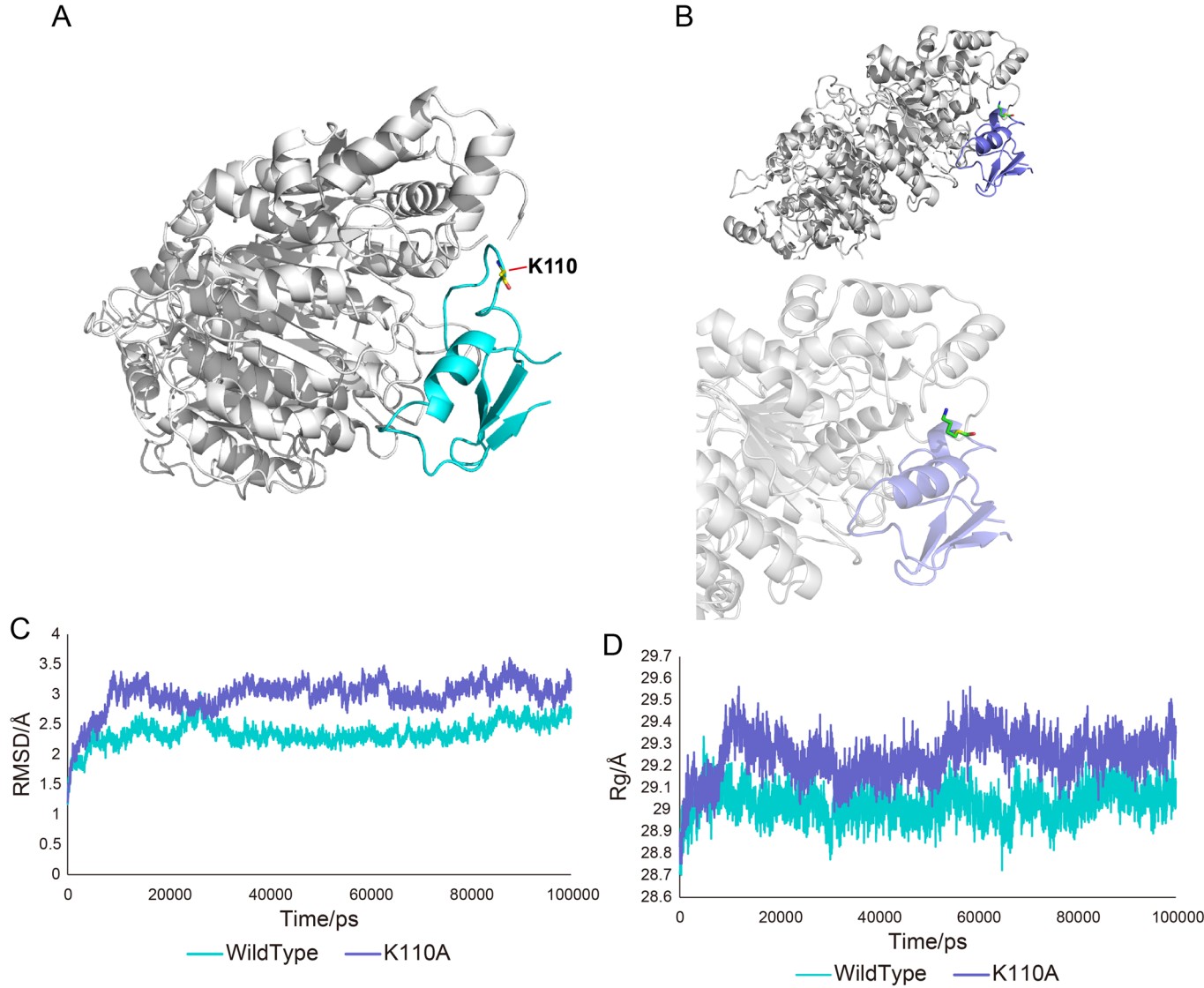

**Figure EV4. TRIM56 positively regulates IFN production.**

(A, B) Models depicting the predictable binding between HDAC6 CD1-CD2 (grey) and TRIM56 B box1 domain (cyan) (F) or TRIM56 K110A (G). (C, D) The plot comparing root mean square deviation (RMSD) (K) or Radius of gyration (Rg) (L) versus simulation time (ps) for WT and K110A mutant. Source data are available online for this figure.

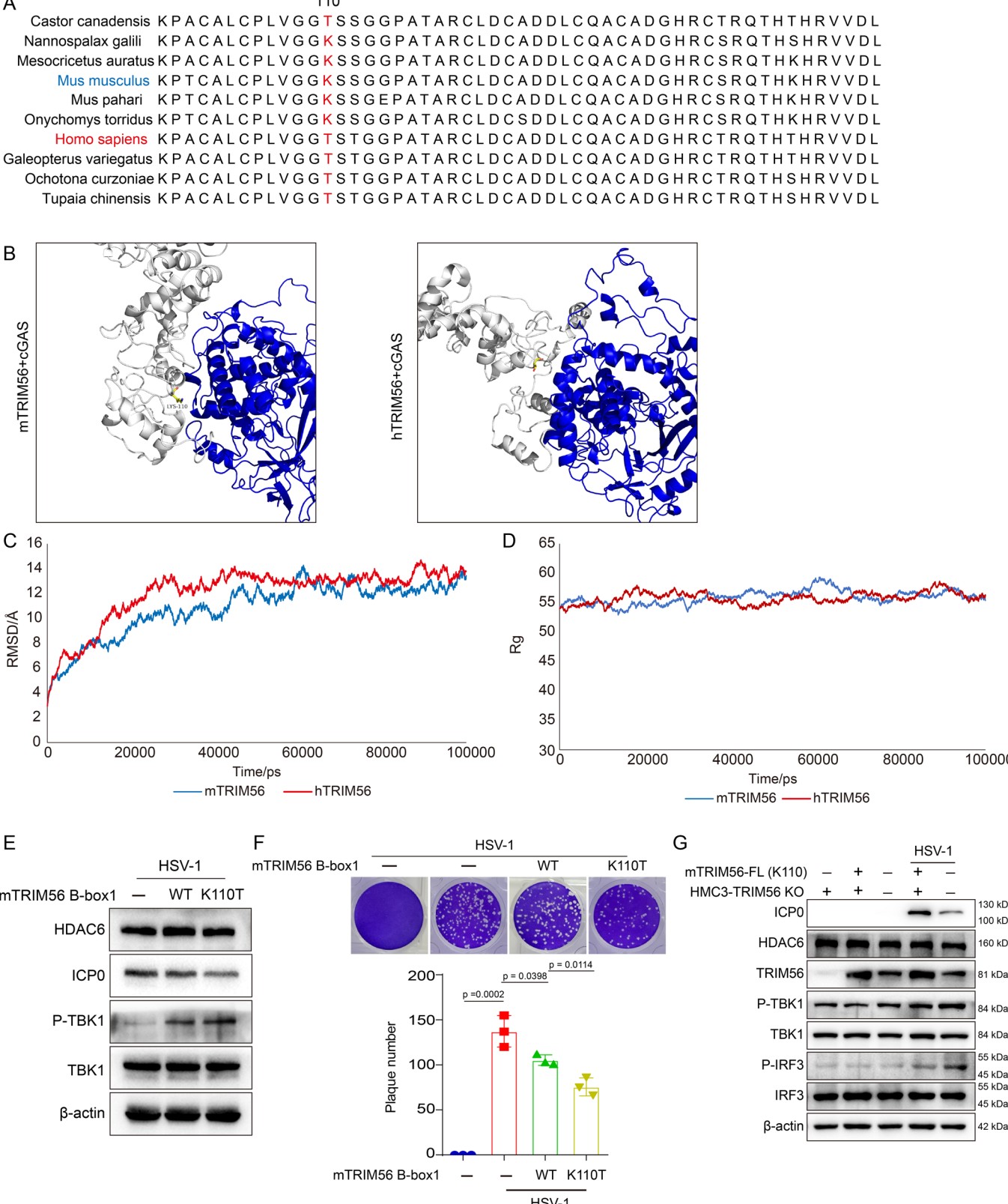

◀    **Figure EV5.  Species-specific amino acids at position 110 of TRIM56 K110 lead to different IFN responses.**

(A) Alignment of amino acids 98–149 of TRIM56 B-box1 from different species. (B) Molecular docking conformation of murine full-length TRIM56 K110 (mTRIM56) or K110T human full-length TRIM56 T110 (hTRIM56) (grey) and cGAS crystals (blue). (C) Plot of root mean square deviation (RMSD) versus simulation time (ps) for mTRIM56 and hTRIM56. (D) Rg comparison across the 100 ns molecular dynamic simulation of mTRIM56 and hTRIM56. (E) N2a cells transfected with mTRIM56 B-box1 or B-box1 K110T for 72 h followed by infection with HSV-1 (MOI = 3) for 12 h, and then western blot analysis of the cGAS-STING signaling pathway. (F) Viral plaque assays of BV2 cells transfected with mTRIM56 or mTRIM56 K110T followed by infection with HSV-1 (MOI = 3) for 12 h. The number of plaques was calculated (right). Data were analyzed using the unpaired *t* test, which are shown as mean ± SD (*n* = 3 biological replicates). (G) Western blot analysis of TBK1 and IRF3 activation in HMC3 TRIM56-KO12 cells transfected with mTRIM56 and then infected with HSV-1 (MOI = 3) for 12 h. Source data are available online for this figure.

