## [Peer Review File · EMBO Reports]

HDAC6 Deacetylates TRIM56 to Negatively Regulate the cGAS-STING-mediated Type I Interferon Response

Yifei Wang, Qiongzen Zeng, Zixin Chen, Shan Li, Ziwei Huang, Zhe Ren, Cuifang Ye, Xiao Wang, Jun Zhou, Kaisheng Liu, and Kai Zheng

Corresponding authors: Yifei Wang (twangyf@jnu.edu.cn), Kai Zheng (zhengk@szu.edu.cn), Kaisheng Liu (liukaisheng@szhospital.com)

Review Timeline:

Submission Date:	12th Feb 24
Editorial Decision:	25th Mar 24
Revision Received:	10th Aug 24
Editorial Decision:	1st Oct 24
Revision Received:	27th Nov 24
Accepted:	13th Dec 24

Editor: Achim Breiling

Transaction Report:

Dear Prof. Wang,

Thank you for the submission of your manuscript to EMBO reports. I have now received the reports from two of the three referees that were asked to evaluate your study, which can be found at the end of this email. A third referee had agreed to assess your study, but has not provided a report, despite repeated chasers. I therefore decided to proceed without his/her input.

As you will see, both referees have several comments, concerns, and suggestions, indicating that a major revision of the manuscript is necessary to allow publication of the study in EMBO reports. As the reports are below, and all the concerns need to be addressed, I will not detail them here.

Given the constructive referee comments, I would like to invite you to revise your manuscript with the understanding that the concerns of the referees must be addressed in the revised manuscript and in a detailed point-by-point response. Acceptance of your manuscript will depend on a positive outcome of a second round of review. It is EMBO reports policy to allow a single round of revision only and acceptance of the manuscript will therefore depend on the completeness of your responses included in the next, final version of the manuscript.

- 1) a .docx formatted version of the final manuscript text (including legends for main figures, EV figures and tables), but without the figures included. Figure legends should be compiled at the end of the manuscript text.
- 2) individual production quality figure files as .eps, .tif, .jpg (one file per figure), of main figures and EV figures. Please upload these as separate, individual files upon re-submission.

- 4) a complete author checklist, which you can download from our author guidelines

(<https://www.embopress.org/page/journal/14693178/authorguide>). Please insert page numbers in the checklist to indicate where the requested information can be found in the manuscript. The completed author checklist will also be part of the RPF.

5) that primary datasets produced in this study (e.g. RNA-seq, CHIP-seq, structural and array data) are deposited in an appropriate public database. If no primary datasets have been deposited, please also state this in a dedicated section (e.g. 'No primary datasets have been generated and deposited'), see below.

The accession numbers and database should be listed in a formal "Data Availability" section (placed after Materials & Methods) that follows the model below. This is now mandatory (like the COI statement). Please note that the Data Availability Section is restricted to new primary data that are part of this study. This section is mandatory. As indicated above, if no primary datasets have been deposited, please state this in this section

Data availability

8) Regarding data quantification and statistics, please make sure that the number "n" for how many independent experiments were performed, their nature (biological versus technical replicates), the bars and error bars (e.g. SEM, SD) and the test used to calculate p-values is indicated in the respective figure legends (also for EV figures and all those in an Appendix). Please also check that all the p-values are explained in the legend, and that these fit to those shown in the figure. Please provide statistical testing where applicable. Please avoid the phrase 'independent experiment', but clearly state if these were biological or technical replicates. Please also indicate (e.g. with n.s.) if testing was performed, but the differences are not significant. In case n=2, please show the data as separate datapoints without error bars and statistics. See also: <http://www.embopress.org/page/journal/14693178/authorguide#statisticalanalysis>

9) Please add scale bars of similar style and thickness to microscopic images, using clearly visible black or white bars (depending on the background). Please place these in the lower right corner of the images themselves. Please do not write on or near the bars in the image but define the size in the respective figure legend.

10) Please also note our reference format:

12) We now use CRediT to specify the contributions of each author in the journal submission system. CRediT replaces the author contribution section. Please use the free text box to provide more detailed descriptions and do not provide your final manuscript text file with an author contributions section. See also our guide to authors: <https://www.embopress.org/page/journal/14693178/authorguide#authorshipguidelines>

13) We would encourage you to use 'Structured Methods', our new Materials and Methods format. According to this format, the Materials and Methods section should include a Reagents and Tools Table (listing key reagents, experimental models, software, and relevant equipment and including their sources and relevant identifiers), uploaded as separate file, followed by a Methods and Protocols section in which we encourage the authors to describe their methods using a step-by-step protocol format with bullet points, to facilitate the adoption of the methodologies across labs. More information on how to adhere to this format as well as downloadable templates (.doc or .xls) for the Reagents and Tools Table can be found in our author guidelines (section 'Structured Methods'):

14) Please order the manuscript sections like this, using these names:

Title page - Abstract - Keywords - Introduction - Results - Discussion - Methods - Data availability section - Acknowledgements - Disclosure and Competing Interests Statement - References - Figure legends - Expanded View Figure legends

I look forward to seeing a revised version of your manuscript when it is ready. Please let me know if you have questions or comments regarding the revision.

Yours sincerely,

Referee #1:

The authors investigated the function of HDAC6 during HSV-1 infection and studied the mechanisms underlying HDAC6-dependent modulation of cGAS/STING activation and antiviral defense in detail, in vitro and in vivo. Through these studies the authors identified that HDAC6 deacetylates TRIM56 which impairs full cGAS activation and antiviral response. This work identified novel mechanistic details of the antiviral cellular response relevant for DNA virus infection. Overall, this impressive work has dissected the aforementioned mechanisms in great detail. The main challenge for the reader is the incomplete description of the performed experiments and results. A few points, listed below, should be addressed.

Major points:

1. A strength of this work is that the authors used a combination of in vitro cell models to study the role of HDAC6 during HSV-1 infection. It is, however, not clear to the reader why the authors decided to use different cell types (neurons and microglia). In addition, given the well-known cell-type specific effects of HSV-1 infection, possible cell-type specific effects, or the lack thereof, should be discussed.

In the discussion, the authors state "Herein, we found that HSV-1 US3 stimulates HDAC6 deacetylase activity to inhibit TRIM56 mediated cGAS/STING IFN activation in microglia and neurons, further implying that HSV 1 utilizes cell specific strategies to counteract the cGAS/STING signaling.". It is not clear how this conclusion is supported by their data.

2. The authors identified very relevant species-dependent differences. It would hence be of importance to introduce the model systems used in detail (BV2, N2a - mouse origin; HMC3 - human origin) and discuss potential differences throughout the manuscript.

3. The term "significant" should only be used where significance is shown. This does not apply to for example Figs. S2H, 5F, 8D, 8E, and 8F.

4. In Fig. 1B, the described increase in pTBK1 and pIRF3 is not clearly visible. The authors should provide quantifications to underpin their claim. In addition, TBK1 shows 6 bands while there are 5 samples. It appears that si-HDAC6 siRNA #2 clearly has a stronger effect in some of the experiments (for example Fig. 1B - ISD) and has been selected for downstream experiments (for example Fig. 1D); this should be discussed.

5. As mentioned in the summary above, some of the approaches and results are incompletely described. For example, the authors state that "We verified the binding interaction between the B-box1 domain of TRIM56 and the CD1-CD2 structural domain of HDAC6 (Fig. 5I)". It is not clear to the reader how this has been verified. The same applies to for example Fig. 5J and Fig. S8.

6. The authors assessed nuclear-cytoplasmic transport of HDAC6 during early HSV-1 infection is not convincingly shown (Fig. 8C and D). The imaging (Fig. 8C) should be repeated using confocal microscopy which will enable to distinguish between nuclear and cytoplasmic localization. In addition, the subcellular fractionation needs to be added to the methods section.

7. Parts of the discussion are extended repetitions of the results and are therefore redundant and can be removed. E.g. paragraph starting with "Our results show that TRIM56 specific acetylation at the K110 residue ...".

Minor points:

8. For non-virologists, more information on HSV-1 pathogenesis and pathology might help to better understand some of the results.

9. The manuscript would benefit from language editing.

10. To make it easier for the reader, the gene and protein nomenclature should be double-checked.

11. The authors describe Fig.S3D with "...H&E stain showed a marked decline in inflammation and cellular damage in the brain...". This should be further explained or indicated in the figure or figure legend.

Referee #2:

Zeng et al., report the finding that HDAC6-mediated deacetylation of K110 located in the so far uncharacterized B-box of TRIM56 suppressed the IFN response by dampening the activity of the DNA sensor cGAS. Mechanistically, acetylated TRIM56 stimulates cGAS activity by monoubiquitination. Interference with this pathway in the context of HSV-1 infection showed an enhanced IFN induction, which suppressed viral replication and reduced infection-induced pathology in mice. The authors show that the viral kinase US3 of HSV-1 activates HDAC6 to further dampen the innate sensing of the viral replication intermediates during early phases of infection.

The study presents biochemical evidence supporting most of their hypothesis. It sheds light on the so far unknown link between TRIM56-mediated regulation of cGAS activity. Nevertheless, some points need clarification and some analysis need to be refined to fully support all claims.

1. The role of US3 mediated HDAC6 activation is unclear to me. The authors cite studies showing that US3 stimulated an antiviral response "The virus protein US3 phosphokinase was found to be a positive regulator of IFN (Wang et al, 2013)" In line with this they claim increased pIRF3 and pTBK1 upon overexpression of US3 (Figure S9C - btw here only pIRF3 is induced, I fail to see increased pTBK1 - see also comment 4). Why would a viral protein promote antiviral immunity? This also contrasts to their discussion and their main conclusion that US3 stimulates the interaction between HDAC and TRIM56 leading to decreased monoubiquitination and dampened cGAS activity.

2. As the authors cite published evidence that US3 interacts / modulates activity of beta-catenin, TBK1 and IRF3 to suppress IFN production. How do the authors biochemically discriminate the observed effect of US3 on HDAC6 from effects of US3 on these downstream proteins? Can US3 overexpression dampen the IFN response in HDAC6 knockdown / knockout cells? How is the viral load of an US3-deficient virus upon infection of HDAC6 knockdown knockout cells?

3. TRIM56 also stimulates STING activation by targeting K150 (PMID: 21074459) How specific the observed effect of TRIM56 effect is for cGAS? Can the authors determine cGAMP levels to show direct effect on cGAS activity? Can the authors investigate the state of STING K150 modification in their system and compare cGAMP levels different knockouts (STING vs cGAS KO) after HSV-1 infection and HDAC6 modulation?

4. All claims on an increase in phosphorylation of signaling molecules that are based on Western blots need quantification and if not available repetitions. Most of the blots the differences are not convincing. e.g. Figure 6B: K110R looks the same at WT = no increase and like state "decreased". Figure 6C: the pull-down of TRIM56 and cGAS western is not convincing due to strong unspecific signal. I would like to ask the authors to critically review their blots across the whole manuscript on whether they look really convincing in regard to their claims or repeat the experiments.

5. Figure 8: The claim on inside-out translocation of HDAC6 shown in panel C is not convincing and does not correlate with the pull-down data in at the 4hrs time point in panel 8D. In the IF panel in C, specifically in the inset of 4h, much more signal is detected in the area of the nucleus, while there is almost no band in the pull-down of the nuclear extract at the same time point. Individual panels of each channel for each timepoint are needed to be clearer. How many times was this performed, could the authors provide more data on this? How do nuclear export inhibitors affect the HDAC6-mediated regulation of cGAS during HSV-1 infection?

Dear editor,

We sincerely appreciate the reviewers for their suggestions concerning our article. As suggested, we carefully addressed the issues raised in the comments. All the revisions were highlighted by colored text. Below are the detailed responses to the reviewer's comments.

Responses to the reviewer #1:

1. A strength of this work is that the authors used a combination of in vitro cell models to study the role of HDAC6 during HSV-1 infection. It is, however, not clear to the reader why the authors decided to use different cell types (neurons and microglia). In addition, given the well-known cell-type specific effects of HSV-1 infection, possible cell-type specific effects, or the lack thereof, should be discussed.

Response: As suggested, additional information regarding the cell-type-specific consequences of HSV-1 infection have been integrated into the discussion section (line 461-470).

2. In the discussion, the authors state "Herein, we found that HSV-1 US3 stimulates HDAC6 deacetylase activity to inhibit TRIM56 mediated cGAS/STING IFN activation in microglia and neurons, further implying that HSV 1 utilizes cell specific strategies to counteract the cGAS/STING signaling.". It is not clear how this conclusion is supported by their data.

Response: As suggested, we have reorganized these descriptions to avoid any misunderstanding in the revised manuscript (line 535-549).

3. The authors identified very relevant species-dependent differences. It would hence be of importance to introduce the model systems used in detail (BV2, N2a - mouse origin; HMC3 - human origin) and discuss potential differences throughout the manuscript.

Response: As suggested, additional information on species-dependent differences has been added in the revised manuscript (line 554-565).

4. The term "significant" should only be used where significance is shown. This does not apply to for example Figs. S2H, 5F, 8D, 8E, and 8F.

Response: As suggested, the term "significant" has been carefully used only where the difference is very obvious.

5. In Fig. 1B, the described increase in pTBK1 and pIRF3 is not clearly visible. The authors should provide quantifications to underpin their claim. In addition, TBK1 shows 6 bands while there are 5 samples. It appears that si-HDAC6 siRNA #2 clearly has a stronger effect in some of the experiments (for example Fig. 1B - ISD) and has been selected for downstream experiments (for example Fig. 1D); this should be discussed.

Response: As suggested, the pTBK1 and pIRF3 bands were quantified in three independent experiments, clearly showing the increase in pTBK1 and pIRF3. Indeed, the knockdown of HDAC6 by si-HDAC6 siRNA #2 was found to be the most effective, and the acetylation modification of its classical downstream protein α -tubulin also demonstrated the greatest degree of change. Furthermore, the best functional regulation of si-HDAC6 siRNA #2 was observed in the investigation of the cGAS-STING signaling pathway. Accordingly, si-HDAC6 siRNA #2 was used for downstream experiments. This information has been added in the revised manuscript (line 102-104)

6. As mentioned in the summary above, some of the approaches and results are incompletely described. For example, the authors state that "We verified the binding interaction between the B-box1 domain of TRIM56 and the CD1-CD2 structural domain of HDAC6 (Fig. 5I)". It is not clear to the reader how this has been verified. The same applies to for example Fig. 5J and Fig. S8.

Response: As suggested, the descriptions have been reorganized in order to provide a more comprehensive and detailed explanation.

7. The authors assessed nuclear-cytoplasmic transport of HDAC6 during early HSV-1 infection is not convincingly shown (Fig. 8C and D). The imaging (Fig. 8C) should be repeated using confocal microscopy which will enable to distinguish between nuclear and cytoplasmic localization. In addition, the subcellular fractionation needs to be added to the methods section.

Response: As recommended, the confocal experiment was repeated using a more suitable HDAC6 antibody (PS02180S, Abmart) for IF experiments (Figure 8C). This revealed that HDAC6 aggregates are present in the perinuclear area at 4 h post-infection. Furthermore, the localization of HDAC6 in the early stages of HSV-1 infection was confirmed by subcellular fractionation assay and quantitative analysis from three independent experiments. In addition, we have incorporated the preparation of nuclear and cytoplasmic extracts into the materials and methods section (line 717-723).

8. Parts of the discussion are extended repetitions of the results and are therefore redundant and can be removed. E.g. paragraph starting with "Our results show that TRIM56 specific acetylation at the K110 residue ...".

Response: As suggested, this section has been reorganized and several repetitions of the results have been removed. In addition, the entire discussion section has been carefully examined to prevent any unnecessary reiteration of the description of the experimental results.

9. For non-virologists, more information on HSV-1 pathogenesis and pathology might help to better understand some of the results.

Response: As suggested, additional information on HSV-1 pathogenesis and pathology has been incorporated into the introduction section (line 19-31 in the revised manuscript).

10. The manuscript would benefit from language editing.

Response: As suggested, we had a native speaker check the grammar of our article to avoid any language or spelling mistakes.

11. To make it easier for the reader, the gene and protein nomenclature should be double-checked.

Response: As suggested, we have carefully checked the gene and protein nomenclature in the revised manuscript.

12. The authors describe Fig.S3D with "...H&E stain showed a marked decline in inflammation and cellular damage in the brain...". This should be further explained or indicated in the figure or figure legend.

Response: As suggested, the detailed information has been added in the figure legend of Appendix Fig. S1D.

Responses to the reviewer #2:

1. The role of US3 mediated HDAC6 activation is unclear to me. The authors cite studies showing that US3 stimulated an antiviral response "The virus protein US3 phosphokinase was found to be a positive regulator of IFN (Wang et al, 2013)" In line with this they claim increased pIRF3 and pTBK1 upon overexpression of US3 (Figure S9C - btw here only pIRF3 is induced, I fail to see increased pTBK1 - see also comment 4). Why would a viral protein promote antiviral immunity? This also contrasts to their discussion and their main conclusion that US3 stimulates the interaction between HDAC and TRIM56 leading to decreased monoubiquitination and dampened cGAS activity.

Response: We are grateful to you for highlighting this crucial issue. In our previous manuscript, we incorrectly stated that US3 was a positive regulator. We offer our sincere apologies for any confusion this may have caused. In fact, HSV-1 US3 functions as a negative IFN regulator, phosphorylating RIG-I and IRF3 to suppress

IFN production (PMID: 24049179; PMID: 34935440). In particular, US3 has been observed to hyperphosphorylate IRF3 and RIG-I, thereby maintaining them in a signaling-repressed state and reducing IFN production. This is consistent with the results of our study, which demonstrated that exogenous overexpression of US3 only increases IRF3 phosphorylation, without any influence on TBK1 (Appendix Fig. S4C). In our study, we demonstrated that US3 phosphorylates HDAC6 to increase its interaction with TRIM56, resulting in the inhibition of cGAS-mediated IFN production. This suggests that US3 utilizes HDAC6-mediated deacetylation as an additional strategy to counter the host antiviral immune response. The description of the relevant errors has been corrected in the revised manuscript.

2. How do the authors biochemically discriminate the observed effect of US3 on HDAC6 from effects of US3 on these downstream proteins? Can US3 overexpression dampen the IFN response in HDAC6 knockdown / knockout cells? How is the viral load of an US3-deficient virus upon infection of HDAC6 knockdown knockout cells?

Response: As suggested, we examined the impact of US3 on the IFN response in HDAC6 knockdown cells. The results demonstrated that US3 significantly reduced the expression of IFN-I during HSV-1 infection. However, the inhibitory effect of US3 was attenuated by the knockdown of HDAC6 (Figure 8K). Thus, US3 exerts its negative effect on IFN-I production, at least in part, through HDAC6.

3. TRIM56 also stimulates STING activation by targeting K150 (PMID: 21074459) How specific the observed effect of TRIM56 effect is for cGAS? Can the authors determine cGAMP levels to show direct effect on cGAS activity? Can the authors investigate the state of STING K150 modification in their system and compare cGAMP levels different knockouts (STING vs cGAS KO) after HSV-1 infection and HDAC6 modulation?

Response: We are in full agreement with this proposition, and an examination of the influence exerted by TRIM56 on STING K150 has the potential to facilitate our understanding of the regulation of antiviral innate immunity by HDAC6 and TRIM56

acetylation. However, these experiments would be considerably more extensive and time-consuming in the context of our current study. In future research, we intend to conduct more comprehensive examinations of STING K150 modification and cGAS in the context of TRIM56 K110 functional exploration. Furthermore, it has been demonstrated that cGAS dimerisation enhances DNA-binding activity and induces a conformational change in cGAS bound to DNA, thereby facilitating the catalysis of cGAMP synthesis (PMID: 24462292, PMID: 24332030). Accordingly, experiments were conducted to ascertain whether TRIM56 K110 acetylation enhances cGAS DNA-binding activity.

4. All claims on an increase in phosphorylation of signaling molecules that are based on Western blots need quantification and if not available repetitions. I would like to ask the authors to critically review their blots across the whole manuscript on whether they look really convincing in regard to their claims or repeat the experiments.

Response: As suggested, the pTBK1 and pIRF3 bands were quantified from three independent experiments in all WB experiments, which clearly demonstrated the discrepancy.

5. Figure 8: The claim on inside-out translocation of HDAC6 shown in panel C is not convincing and does not correlate with the pull-down data in at the 4hrs time point in panel 8D. In the IF panel in C, specifically in the inset of 4h , much more signal is detected in the area of the nucleus, while there is almost no band in the pull-down of the nuclear extract at the same time point. Individual panels of each channel for each timepoint are needed to be clearer. How many times was this performed, could the authors provide more data on this? How do nuclear export inhibitors affect the HDAC6-mediated regulation of cGAS during HSV-1 infection?

Response: As suggested, the confocal experiment was repeated using a more suitable HDAC6 antibody (PS02180S, Abmart) for IF experiments (Figure 8C). This revealed that HDAC6 aggregates are present in the perinuclear area at 4 h post-infection.

Furthermore, the localization of HDAC6 in the early stages of HSV-1 infection was confirmed by subcellular fractionation assay and quantitative analysis from three independent experiments (Figure 8D).

Responses to the reviewer #3:

1. The quality of several western blots is not sufficient to draw a conclusion.

Response: As suggested, the WB bands were quantified from three independent experiments in all WB experiments.

2. No direct biochemical evidence is provided that HDAC6 deacetylates TRIM56 at k110.

Response: As shown in Figure 4L, N2a cells were infected with HSV-1 following the overexpression of Flag-TRIM56-WT or Flag-TRIM56-K110R. The CO-IP Flag analysis demonstrated a notable elevation in the acetylation of Flag-TRIM56-K110R in comparison to Flag-TRIM56-WT. Furthermore, the acetylation modification level of Flag-TRIM56-WT was markedly diminished when co-transfected with HDAC6. Conversely, the acetylation level of Flag-TRIM56-K110R exhibited minimal alteration, indicating that the acetylation of TRIM56 K110 was markedly regulated by HDAC6 deacetylase activity.

3. For the statistical analysis, in several cases the authors use SEM (which is used to compare the mean/average of large samples), where they should in fact use SD (which is used to show the experimental variation)

Response: The majority of the statistical data presented in this text has been re-analyzed, and the corresponding figure notes have been updated to reflect the resulting SD value.

4. Figure 1B, enhanced activation of TBK and IRF3 in HDAC6 Knock-down is not obvious under HSV-1 infection. A quantification is necessary.

Response: As suggested, the relative phosphorylation levels of TBK1 and IRF3 have

been quantified and are presented in Figure 1A-1C.

5. In Figure 2D, siRNA treatment can induce IFN- β release at 0 hr infection, is it normal? (meaning KD HDAC6 alone is enough to induce IFN β release, which seems strange)

Response: As suggested, the experimental analyses were repeated and the results are presented in Figure 1E. These results demonstrate that there was no significant upregulation of IFN β at 0 hours of HSV-1 infection following siHDAC6 knockdown.

6. Figure 2A, the plaque assay results are not really convincing; multiple timepoints would make this more clear.

Response: As suggested, this assay was conducted at multiple time points. N2a cells transfected with HDAC6 siRNA were infected with HSV-1 for 0 h, 4 h, 8 h, and 12 h. Subsequently, an indirect viral plaque assay was performed on Vero cells, and statistical analysis revealed that siRNA-mediated HDAC6 knockdown resulted in a time-dependent inhibition of viral plaque formation (Fig 2A in the revised manuscript).

7. Figure 4G is wrong. The description is "By co-immunoprecipitation assay, we found that HSV 1 infection enhanced the interaction between HDAC6 and TRIM56 in N2a cell s (Fig 4G)". But in the figure, the interaction is lost between HDAC6 and Trim56 under HSV infection.

Response: We thank the reviewer for pointing out this error and we have changed the description to “By co-immunoprecipitation assay, we found that HSV-1 infection reduced the interaction between HDAC6 and TRIM56 in N2a cells (Fig 4G)”.

8. Figure 4I, the western is terrible. Signal of Acetyllysine is not really visible (same as Figure 4L, HDAC6 and TRIM56 signal)

Response: To obtain more obvious outcomes, the quantity of protein used in the CO-IP assay was increased, and the degree of acetylation modification of TRIM56 by

CO-IP TRIM56 was revalidated.

9. For Figure 5, experiment with expression of FL Trim56 is needed. Otherwise, as a E2, only a B-box domain is not sufficient to maintain the ub ligase function.

Response: As suggested, a comparison experiment between FL-TRIM56 and B-box1 was incorporated into Figure 5B, Figure 5F and 5G, which demonstrated that FL-TRIM56 was more effective than B-box1 in enhancing IFN-I production. Furthermore, the immunoprecipitation results demonstrated that FL-TRIM56 markedly elevated the extent of cGAS monoubiquitination, indicating that B-box1 may serve as a pivotal structural domain that amplifies TRIM56's regulatory role in cGAS monoubiquitination and IFN-I.

10. in Section "The acetylation of TRIM56 K110 is critical for its binding to HDAC6 and regulation of the IFN response". Author did some TRIM56 KD experiments, and said "...Therefore, these results further supported the hypothesis that HDAC6 modulates the TRIM56 mediated antiviral response via deacetylation". The experiment before this statement is TRIM56 KD by siRNA and has nothing to do with HDAC6. So, the conclusion is not correct.

Response: As suggested, we have changed the statement "...Therefore, these results further supported the hypothesis that HDAC6 modulates the TRIM56 mediated antiviral response via deacetylation" to "Therefore, these results further supported that TRIM56 is a positive IFN regulator."

11. In Figure 5, how to explain the group No.3, which is reduced in Fig5E, while in Fig 5C, it has no effect on mRNA level?

Response: Figure 5E and Figure 5C represent two independent experiments. The lack of significant difference observed in group No. 3 of Figure 5C may be attributed to the large error associated with the duplicate experiments. To further confirm the knockdown efficiency of ASO-TRIM56 B-box1, additional 5 independent duplicate experiments were conducted, and RT-qPCR statistical analysis demonstrated that all

three ASO-TRIM56 B-box1 resulted in a significant knockdown of TRIM56 mRNA levels.

12. Critical point: Figure 5, Molecular docking is not a verification. It can be used as supporting evidence, but not to demonstrate that the two proteins bind. A real biochemical experiment with purified proteins should be done (if the authors want to make this point).

Response: We agree with this comment that molecular docking is not a verification. However, we have confirmed the interactions between HDAC6 and TRIM56 by immunoprecipitation in Figure 4 (Figure 4G-4K) and the regulation of TRIM56 K110 acetylation by HDAC6 (Figure 4L).

13. In Figure 6, for a TRIM56 IP, TRIM56 should be enriched so that its signal in IB part should be very strong; how could it be so weak??? The signal is stronger in the input...

Response: As illustrated in Figure 4I, to obtain more obvious outcomes, the quantity of protein used in the CO-IP assay was increased, resulting in the upregulation of TRIM56 intensity in the IB. Moreover, to substantiate the hypothesis that HDAC6 can influence the binding of TRIM56 to cGAS and the ubiquitination of cGAS, a CO-IP cGAS assay was performed, which clearly indicated that both the TRIM56-cGAS interaction and the ubiquitination of cGAS are affected by the deacetylase activity of HDAC6 (Figure 6C).

14. Figure 6 C, I did not see an inhibition on cGAS-Trim56 interaction by expressing FL HDAC6. In the control group, I cannot see that cGAS is captured by TRIM56 IP.

Response: As illustrated in Figure 6C of the revised manuscript, we conducted a repetition of the CO-IP assay. Following the overexpression of FL-HDAC6 and HDAC6 HD1/2m, a CO-IP was conducted to detect the interactions between cGAS and TRIM56, as well as the level of cGAS ubiquitination modification. The results of the immunoprecipitation indicated that the overexpression of HDAC6 significantly

inhibited TRIM56 binding to cGAS and reduced cGAS monoubiquitination.

15. Figure 6G, for visualizing mono-ubiquitinated cGAS, IB-Ubiquitin is not a proper way to show it. More evidence is needed to argue that the indicated band is a mono-ubiquitinated cGAS. Furthermore, all the western blots were missing the molecule weight/protein marker labelling, making it more difficult for reviewers to analyze the western blot result.

Response: Previous studies have confirmed the impact of TRIM56 on the conversion of cGAS ubiquitination to monoubiquitination (PMID: 29426904). In our study, we did not conduct additional experiments to confirm the monoubiquitination of cGAS; however, we validated this through co-immunoprecipitation of cGAS and analysis using the monoubiquitination-specific VU-1 antibody. The results revealed that ubiquitination bands on the immunoblots membrane were only observed in the protein marker range of 70 kDa to 55 kDa above the heavy chain of the antibody. Furthermore, molecular weight/protein marker labeling was incorporated into the western blots.

16. The mice work is missing TRIM56 K110R data; this would nicely complement what is presented.

Response: We are in deep agreement with this suggestion that the results from the TRIM56 K110R mice would strongly support our conclusion. It is possible that in future work we will carry out more detailed studies in the TRIM56 K110R mice.

17. Figure 8C, IF data about HDAC6. It is very strange that HDAC6 is in the nucleus. If the authors want to prove their findings, a quantification of nucleus-HDAC6 and cytosolic-HDAC6 signal would be useful. The HDAC6 antibody (D21B10) from CST is not recommended for IF. The authors should try another HDAC6 specific antibody.

Response: As suggested, the confocal experiment was repeated using a more suitable HDAC6 antibody (PS02180S, Abmart) for IF experiments (Figure 8C). This revealed that HDAC6 aggregates are present in the perinuclear area at 4 h post-infection.

Furthermore, the localization of HDAC6 in the early stages of HSV-1 infection was confirmed by subcellular fractionation assay and quantitative analysis from three independent experiments.

18. Quantification in figure 2 B and C used SEM, please change it to SD.

Response: As requested, we have re-quantified the results of figure 2 B and C and have changed the SEM to SD.

19. Method: Authors used lentivirus to create HSE model in mice.

Response: In fact, the BALB/c mouse intranasally infected with HSV-1 is used to establish the HSE model. In addition, the BALB/c mouse with brain overexpression of TRIM56 using lentivirus was established as our previous work (PMID: 34902449).

20. Statistical analysis in Figure 7 E to H is missing.

Response: As requested, significant difference analyses were performed using the unpaired t-test for Figure 7 E to H.

21. Figure 8I, Phosphoserine band at 4 hr is missing

Response: The shorter exposure time resulted in the observation of lighter serine phosphorylation bands of HDAC6 at the 4hr. To validate changes in HDAC6 phosphorylation levels at the early time points of HSV-1 infection, we re-performed the immunoprecipitation assay, as illustrated in Figure 8I.

Again, we appreciate you and the reviewers for the time and suggestions on our manuscript. We are looking forward to hearing from you and remain available for any inquiry.

Sincerely,

Yifei Wang

Institute of Biomedicine, College of Life Science and Technology, Jinan University,
Guangzhou 510632, China.

Dear Prof. Wang,

Thank you for the submission of your revised manuscript to our editorial offices. I have now received the reports from the three referees that I asked to re-evaluate the study, you will find below. As you will see, the referees now support the publication of the study in EMBO reports. However, all three referees have remaining concerns or suggestions to improve the manuscript, I ask you to address in a final revised manuscript. Please also provide a final p-b-p-response addressing these points.

Regarding point three of referee #2, I agree that the experiments proposed are out of scope of the present study. No further response is needed.

Moreover, I strongly support the suggestion of referee #3 to re-organize the manuscript, moving some more data to the Appendix (see also below). Please do that. However, it is not necessary to label the figures itself.

- Please provide institutional e-mail addresses for all corresponding authors on the title page of the manuscript.
 - We now use CRediT to specify the contributions of each author in the journal submission system. CRediT replaces the author contribution section. Please use the free text box to provide more detailed descriptions and do NOT provide your final manuscript text file with an author contributions section. See also our guide to authors:
<https://www.embopress.org/page/journal/14693178/authorguide#authorshipguidelines>
 - Please order the manuscript sections like this, using these names:
Title page - Abstract - Keywords - Introduction - Results - Discussion - Methods - Data availability section - Acknowledgements - Disclosure and Competing Interests Statement - References - Figure legends - Expanded View Figure legends
 - Please remove the section "Supporting information" from manuscript text file.
 - The data availability section is restricted to datasets deposited at external repositories. Please remove all unrelated information from this section. Moreover, please remove the referee access and make sure that the datasets are public latest upon publication of the study. Please also add a direct and specific URL to reach PXD046808.
 - Please make sure that the number "n" for how many independent experiments were performed, their nature (biological versus technical replicates), the bars and error bars (e.g. SEM, SD) and the test used to calculate p-values is indicated in the respective figure legends. Please also check that all the p-values are explained in the legend, and that these fit to those shown in the figure. Please provide statistical testing where applicable. Please avoid the phrase 'independent experiment', but clearly state if these were biological or technical replicates. Please also indicate (e.g. with n.s.) if testing was performed, but the differences are not significant. In case n=2, please show the data as separate datapoints without error bars and statistics. See also:
<http://www.embopress.org/page/journal/14693178/authorguide#statisticalanalysis>
- If n<5, please show single datapoints for diagrams. Moreover:
- Please define the annotated p values ***/**/* as well as provide the exact p-values for the same in the legend of figure 1a-c; 2c-d; 5b, f; 8h; EV 2a-c; EV 3a; EV 4b; as appropriate.
 - Please provide the exact p values in the legends of figures 1d-i; 2a-b, e-f; 3f-g; 5c, e, i-j; 6a, f-g; 7d-h, j; 8b, g, k; EV 1b, f-h; EV 2d-e; EV 4c-d; EV 5e.
 - Please indicate the statistical test used for data analysis in the legends of figures 1a-c; 2c-d; 4b; 5b, f; 8d, h; EV 2a-c; EV 3a, g; EV 4b.
 - Please note that in figures 1d-e; 8g; EV 1f; EV 2d-e; there is a mismatch between the annotated p values in the figure legend and the annotated p values in the figure file that should be corrected.
 - Please note that information related to n is missing in the legends of figures 5b, f; 8d, h; EV 2a-c; EV 4b.
 - Although 'n' is provided, please describe the nature of entity for 'n' in the legends of figures 1a-i; 2a-f; 5c, e, i-j; 6a, f-g; 8b, g, k; EV 1b, f-h; EV 2d-e; EV 4c-d; EV 5e.
 - Please note that the error bars are not defined in the legends of figures 3e; 5b, f; 8d, h; EV 2a-c; EV 4b.
 - Please note that the black arrowheads are not defined in the legend of figure 4i, 5g, 6b-c, e. This needs to be rectified.
- Please add scale bars of similar style and thickness to all microscopic images (main, EV and Appendix figures), using clearly visible black or white bars (depending on the background). Please place these in the lower right corner of the images themselves. Please do not write on or near the bars in the image but define the size in the respective figure legend. Presently, most scale bars are too thin and/or have text nearby. Please improve.
 - Please make sure that all figure panels are called out separately and sequentially. It seems, presently a callout for panel 1E is missing. Moreover, there is a callout for Fig. EV6E, but there are only 5 EV figures. Please check.

- Please name the Appendix file 'Appendix' and provide it as pdf. Please add a table of content and with page numbers to the title page and remove the author names. Please include all Appendix items in the Appendix file and do not upload them separately. Moreover, please move all the methods information to the main manuscript methods section. We do not allow methods information in the Appendix.

- There are 8 Supplemental tables uploaded. Appendix Tables S1-S3 are datasets. Please upload these as dataset files, renamed to Dataset EV1-EV3 and called out like this. Please add legends for these on the first TAB in each Excel file. Please add present Appendix Tables S4 and S5 to the Appendix file and name this Appendix Table S1 and S2. Please update their callouts. Present Appendix Tables S6-S8 need to be moved to the reagents & tools table (see below).

- All Materials and Methods need to be described in the main text using our 'Structured Methods' format, which is required for all research articles. According to this format, the Materials and Methods section should include a Reagents and Tools Table (listing key reagents, experimental models, software, and relevant equipment and including their sources and relevant identifiers), uploaded as separate file, followed by a Methods section in which we encourage the authors to describe their methods using a step-by-step protocol format with bullet points, to facilitate the adoption of the methodologies across labs. More information on how to adhere to this format as well as downloadable templates (.doc) for the Reagents and Tools Table can be found in our author guidelines (section 'Structured Methods'):

- As I already indicated in my first decision letter, we now request the publication of original source data (SD) with the aim of making primary data more accessible and transparent to the reader. It seems you have been contacted already by our source data coordinator, who indicated which figure panels we would need source data for. The presently provided SD is incomplete. I attach again the source data checklist. Please make sure that all the requested source data is provided. Please upload all source data for one figure saved in a single folder, zipped and then uploaded. For EV and/or Appendix figures, ZIP together all source data can be combined/zipped together into one folder and uploaded. Finally, please upload the filled in checklist with your final revised manuscript.

- During our data integrity screen of the figures, we noted that the b-actin Western blot image in panel 3A (Trigeminus) and 3H (P/M/C) looks very similar, although different tissues are examined (as I understand). This is the same in the related source data. Please check if the correct panels are shown and explain.

In addition, I would need from you uploaded separately:

Best,

Referee #1:

Most of my comments have been adequately addressed by the authors. Some further clarification is needed in Fig. 8 (details below) as well as a few minor comments.

1) The description of Fig. 8C states: "The immunofluorescence assay showed that HDAC6 disseminated in the cytosol during the early stage of HSV-1 infection (< 6 h.p.i) (Fig. 8C)". I am not able to connect the description and the figure (or the figure legend). What is it that the authors actually want to show here? The figure legend mentions F-actin (red) which I don't see. Also, which MOI was used? If these really are confocal pictures, it should be clearly stated.

2) Fig. EV1B needs some adjustment.

3) The authors may consider splitting the section "Deficiency of HDAC6 protects mice against HSV-1 infection" into two since the first half of the paragraph is describing in vitro studies.

4) In line 151 it is written "Hdac6^{-/-} mice were intranasally infected with HSV-1". This may cause confusion as it sounds like only KOs were infected.

Referee #2:

1. Figure S4C in the appendix is not convincing because of the bad quality of the blot (specifically for pTBK1) as technical information is missing on which antibody was used to detect the pSer175 in IRF3. Conceptually, if US3 directly inhibits IRF3, I wonder why the more complicated HDAC6-TRIM56 axis of inhibition also evolved.

2. ok

3. not answered

4. ok

5. ok

Referee #3:

I have now looked carefully at the revised version of the manuscript. While I see that the authors have addressed most of my original comments, I still find the paper rather difficult to follow. I think this is partly because the authors have too many panels in each figure, so that it is difficult to find/follow the key information. A reorganization of the figures layout, with transfer of some material in the supplementary section, might help. I note that the Figures in the revised manuscript are not labelled (e.g. Fig 1, Fig 2,...) and this makes it difficult for the reviewer to know with certainty which Figure is which; this should definitely not be.

Dear editor,

We sincerely appreciate the reviewers for their suggestions concerning our article. As suggested, we carefully addressed the issues raised in the comments. All the revisions were highlighted by colored text. Below are the detailed responses to the reviewer's comments.

Responses to editor:

1. Please provide institutional e-mail addresses for all corresponding authors on the title page of the manuscript.

Response: We checked and revised the institutional email addresses of all corresponding authors on the title page of the manuscript.

2. We now use CRediT to specify the contributions of each author in the journal submission system. CRediT replaces the author contribution section. Please use the free text box to provide more detailed descriptions and do NOT provide your final manuscript text file with an author contributions section. See also our guide to authors: <https://www.embopress.org/page/journal/14693178/authorguide#authorshipguidelines>

Response: We have made the modifications as requested, removing the author contributions section from the manuscript file.

3. Please order the manuscript sections like this, using these names:

Title page - Abstract - Keywords - Introduction - Results - Discussion - Methods - Data availability section - Acknowledgements - Disclosure and Competing Interests Statement - References - Figure legends - Expanded View Figure legends

Response: We have made the modifications as requested, "Availability of data and materials" to "Data availability section", and "Conflicts of interest" to "Disclosure and Competing Interests Statement".

4. Please remove the section "Supporting information" from manuscript text file.

Response: We have removed the section "Supporting information" from manuscript text file.

5. The data availability section is restricted to datasets deposited at external repositories. Please remove all unrelated information from this section. Moreover, please remove the referee access and make sure that the datasets are public latest upon publication of the study. Please also add a direct and specific URL to reach PXD046808.

Response: We have modified the data availability section as required (line 887-891 in the revised manuscript). Since the dataset was uploaded under the "Private" option, the disclosure timing cannot be adjusted during the process. The dataset will be made publicly available immediately after the article's DOI or PMID is obtained.

6. Please make sure that the number "n" for how many independent experiments were performed, their nature (biological versus technical replicates), the bars and error bars (e.g. SEM, SD) and the test used to calculate p-values is indicated in the respective figure legends. Please also check that all the p-values are explained in the legend, and that these fit to those shown in the figure. Please provide statistical testing where applicable. Please avoid the phrase 'independent experiment', but clearly state if these were biological or technical replicates. Please also indicate (e.g. with n.s.) if testing was performed, but the differences are not significant. In case n=2, please show the data as separate datapoints without error bars and statistics. See also:

<http://www.embopress.org/page/journal/14693178/authorguide#statisticalanalysis>

If n<5, please show single datapoints for diagrams. Moreover:

*- Please define the annotated p values ***/**/* as well as provide the exact p-values for the same in the legend of figure 1a-c; 2c-d; 5b, f; 8h; EV 2a-c; EV 3a; EV 4b; as appropriate.*

Response: We have defined the exact p values in figure 1A-C, figure 2C-D, etc.

- Please provide the exact p values in the legends of figures 1d-i; 2a-b, e-f; 3f-g; 5c, e,

i-j; 6a, f-g; 7d-h, j; 8b, g, k; EV 1b, f-h; EV 2d-e; EV 4c-d; EV 5e.

Response: We have defined the exact p values in figure 1D-I, figure 2A-B, figure 2E-F, etc.

- Please indicate the statistical test used for data analysis in the legends of figures 1a-c; 2c-d; 4b; 5b, f; 8d, h; EV 2a-c; EV 3a, g; EV 4b.

Response: We have indicated the statistical test used for data analysis in the legends of figure 1A-C, figure 2C-D, figure 4B, etc.

- Please note that in figures 1d-e; 8g; EV 1f; EV 2d-e; there is a mismatch between the annotated p values in the figure legend and the annotated p values in the figure file that should be corrected.

Response: We have checked and defined the exact p values in figure 1D-E, EV1F, EV2D-E, etc.

- Please note that information related to n is missing in the legends of figures 5b, f; 8d, h; EV 2a-c; EV 4b.

Response: We have checked the full text and supplemented the relevant information about n.

- Although 'n' is provided, please describe the nature of entity for 'n' in the legends of figures 1a-i; 2a-f; 5c, e, i-j; 6a, f-g; 8b, g, k; EV 1b, f-h; EV 2d-e; EV 4c-d; EV 5e.

Response: We have described the nature of entity for 'n' in the legends of figures 1A-I, 2A-F, etc.

- Please note that the error bars are not defined in the legends of figures 3e; 5b, f; 8d, h; EV 2a-c; EV 4b.

Response: We have defined the error bars in the legend of the figures.

- Please note that the black arrowheads are not defined in the legend of figure 4i, 5g, 6b-c, e. This needs to be rectified.

Response: We have defined the black arrow in the legend of the figures.

7.- Please add scale bars of similar style and thickness to all microscopic images (main, EV and Appendix figures), using clearly visible black or white bars (depending on the background). Please place these in the lower right corner of the images

themselves. Please do not write on or near the bars in the image but define the size in the respective figure legend. Presently, most scale bars are too thin and/or have text nearby. Please improve.

Response: We have modified the scale bars of all microscopic images.

8.- Please make sure that all figure panels are called out separately and sequentially. It seems, presently a callout for panel 1E is missing. Moreover, there is a callout for Fig. EV6E, but there are only 5 EV figures. Please check.

Response: Thank you for pointing out the error, which has been corrected. We have also checked the whole manuscript to avoid a similar mistake.

9.- Please name the Appendix file 'Appendix' and provide it as pdf. Please add a table of content and with page numbers to the title page and remove the author names. Please include all Appendix items in the Appendix file and do not upload them separately. Moreover, please move all the methods information to the main manuscript methods section. We do not allow methods information in the Appendix.

Response: We have moved all the methods information from appendix file to the main manuscript methods section. At the same time, it has been modified according to the requirements of the appendix file.

10. There are 8 Supplemental tables uploaded. Appendix Tables S1-S3 are datasets. Please upload these as dataset files, renamed to Dataset EV1-EV3 and called out like this. Please add legends for these on the first TAB in each Excel file. Please add present Appendix Tables S4 and S5 to the Appendix file and name this Appendix Table S1 and S2. Please update their callouts. Present Appendix Tables S6-S8 need to be moved to the reagents & tools table (see below).

Response: We have renamed Appendix tables S1-S3 to data sets EV1-EV3, and Appendix Tables S4 and S5 to Appendix tables S1 and S2. We have modified Appendix tables S6-S8, the relevant content of the genotyping primers of the mouse HDAC6 gene (the original Appendix tables S6) and the siRNA sequences (the original

Appendix tables S8) has been moved to the Methods section, and the PCR primer sequences (the original Appendix tables S7) have been included in "the reagents & tools table".

11. All Materials and Methods need to be described in the main text using our 'Structured Methods' format, which is required for all research articles. According to this format, the Materials and Methods section should include a Reagents and Tools Table (listing key reagents, experimental models, software, and relevant equipment and including their sources and relevant identifiers), uploaded as separate file, followed by a Methods section in which we encourage the authors to describe their methods using a step-by-step protocol format with bullet points, to facilitate the adoption of the methodologies across labs. More information on how to adhere to this format as well as downloadable templates (.doc) for the Reagents and Tools Table can be found in our author guidelines (section 'Structured Methods'):

Response: We have made the modifications as requested.

12. As I already indicated in my first decision letter, we now request the publication of original source data (SD) with the aim of making primary data more accessible and transparent to the reader. It seems you have been contacted already by our source data coordinator, who indicated which figure panels we would need source data for. The presently provided SD is incomplete. I attach again the source data checklist. Please make sure that all the requested source data is provided. Please upload all source data for one figure saved in a single folder, zipped and then uploaded. For EV and/or Appendix figures, ZIP together all source data can be combined/zipped together into one folder and uploaded. Finally, please upload the filled in checklist with your final revised manuscript.

Response: We have organized and compressed the original data of each Figure, including western blot bands, immunofluorescence images, and other data.

13. During our data integrity screen of the figures, we noted that the b-actin Western blot image in panel 3A (Trigeminus) and 3H (P/M/C) looks very similar, although different tissues are examined (as I understand). This is the same in the related source data. Please check if the correct panels are shown and explain.

Response: We thank you for pointing out the mistake in the layout process of Figure 3. We have checked the original data and promptly corrected the β -actin western blot image in Trigeminus (the original panel 3A move to current panel S1A), as well as updated the uploaded source data.

Responses to the reviewer #1:

1. The description of Fig. 8C states: "The immunofluorescence assay showed that HDAC6 disseminated in the cytosol during the early stage of HSV-1 infection (< 6 h.p.i) (Fig. 8C)". I am not able to connect the description and the figure (or the figure legend). What is it that the authors actually want to show here? The figure legend mentions F-actin (red) which I don't see. Also, which MOI was used? If these really are confocal pictures, it should be clearly stated.

Response: We thank the reviewer for pointing out this error in our figure captions and we have made revisions to the figure legend and methods section based on the confocal images (the original figure Figure 8C has been modified to current Figure 8B). The description of Figure 8B is to be rewritten as "The immunofluorescence assay showed that HDAC6 aggregates are present in the perinuclear area at 4 h post-infection (Fig. 8B)"(line 413-414 in the revised manuscript).

2. Fig. EV1B needs some adjustment.

Response: We thank the reviewer for pointing out this error in Figure EV1B and we have made the necessary corrections to Figure EV1B.

3. The authors may consider splitting the section "Deficiency of HDAC6 protects mice against HSV-1 infection" into two since the first half of the paragraph is describing in

vitro studies.

Response: As suggested, we have divided the section into two parts: "Inhibition of HDAC6 restricts HSV-1 infection in vitro" and "Deficiency of HDAC6 protects mice against HSV-1 infection".

4. In line 151 it is written "Hdac6^{-/-} mice were intranasally infected with HSV-1". This may cause confusion as it sounds like only KOs were infected.

Response: We thank the reviewer for pointing out this error and have corrected the statement (line 153-155 in the revised manuscript).

Responses to the reviewer #2:

Figure S4C in the appendix is not convincing because of the bad quality of the blot (specifically for pTBK1) as technical information is missing on which antibody was used to detect the pSer175 in IRF3. Conceptually, if US3 directly inhibits IRF3, I wonder why the more complicated HDAC6-TRIM56 axis of inhibition also evolved.

Response: Thank you very much for this suggestion. Our results in Figure 8 suggests that during the early stages of infection, HSV-1 US3 impedes the onset of the antiviral immune response by regulating the HDAC6-TRIM56 signaling. However, once the antiviral immune response has been initiated and activated (at the mid-viral infection stage), US3 directly hyperphosphorylates and inhibits IRF3, thereby suppressing the antiviral response and maintaining a low level of immunity that favors viral replication.

Responses to the reviewer #3:

I have now looked carefully at the revised version of the manuscript. While I see that the authors have addressed most of my original comments, I still find the paper rather difficult to follow. I think this is partly because the authors have too many panels in each figure, so that it is difficult to find/follow the key information. A reorganization of the figures layout, with transfer of some material in the supplementary section, might help. I note that the Figures in the revised manuscript are not labelled (e.g. Fig

1, Fig 2,...) and this makes it difficult for the reviewer to know with certainty which Figure is which; this should definitely not be.

Response: Thank you very much for the suggestions on our data formatting. In accordance with your advice, the presentation of the charts and tables has been restructured. For instance, the original figures 3A,B, 6I,J, 7E-H, and EV3A-E have been adjusted and are now presented as figures S1A,D, EV5A,G, S4D-G, and S3A-E, respectively.

Again, we appreciate you and the reviewers for the time and suggestions on our manuscript. We are looking forward to hearing from you and remain available for any inquiry.

Sincerely,

Yifei Wang

Institute of Biomedicine, College of Life Science and Technology, Jinan University,
Guangzhou 510632, China.

Prof. Yifei Wang
Jinan University
College of life science and technology
No. 601 Huangpu Road West, Guangzhou
Guangzhou, Guangdong 510632
China

Dear Prof. Wang,

I am very pleased to accept your manuscript for publication in the next available issue of EMBO reports. Thank you for your contribution to our journal.

Yours sincerely,
